# One Stone with Two Birds: A Null-Text-Null Frequency-Aware Diffusion Models for Text-Guided Image Inpainting

**Haipeng Liu**[†]   **Yang Wang**[†][*]   **Meng Wang**
School of Computer Science and Information Engineering
Hefei University of Technology, China
`hpliu_hfut@hotmail.com`  `yangwang@hfut.edu.cn`  `eric.mengwang@gmail.com`

## Abstract

Text-guided image inpainting aims at reconstructing the masked regions as per text prompts, where the longstanding challenges lie in the preservation for unmasked regions, while achieving the semantics consistency between unmasked and inpainted masked regions. Previous arts failed to address both of them, always with either of them to be remedied. Such facts, as we observed, stem from the entanglement of the hybrid (e.g., mid-and-low) frequency bands that encode varied image properties, which exhibit different robustness to text prompts during the denoising process. In this paper, we propose a null-text-null frequency-aware diffusion models, dubbed **NTN-Diff**, for text-guided image inpainting, by decomposing the semantics consistency across masked and unmasked regions into the consistencies as per each frequency band, while preserving the unmasked regions, to circumvent two challenges in a row. Based on the diffusion process, we further divide the denoising process into early (high-level noise) and late (low-level noise) stages, where the mid-and-low frequency bands are disentangled during the denoising process. As observed, the stable mid-frequency band is progressively denoised to be semantically aligned during text-guided denoising process, which, meanwhile, serves as the guidance to the null-text denoising process to denoise low-frequency band for the masked regions, followed by a subsequent text-guided denoising process at late stage, to achieve the semantics consistency for mid-and-low frequency bands across masked and unmasked regions, while preserve the unmasked regions. Extensive experiments validate the superiority of NTN-Diff over the state-of-the-art diffusion models to text-guided diffusion models. Our code can be accessed from `https://github.com/htyjers/NTN-Diff`.

## 1 Introduction

Traditional image inpainting methods [23, 24, 3, 41, 46, 16, 19] primarily rely on unmasked regions as guidance to recover masked areas, which are widely applied to object removal [18, 45, 28] and photo restoration [21, 2, 27]. Based on that, text-guided image inpainting [1, 35, 47, 4, 36, 17] has achieved remarkable progress, offering a more flexible and diverse approach. These methods focus on reconstructing user-specified objects within masked regions given text prompts.

The success of text-guided image inpainting is largely attributed to the powerful generative capabilities of Denoising Diffusion Probabilistic Models (DDPMs) [9], *e.g.*, Stable Diffusion [33]. As a critical

---

† Equal contribution
* Yang Wang is the corresponding author

39th Conference on Neural Information Processing Systems (NeurIPS 2025).

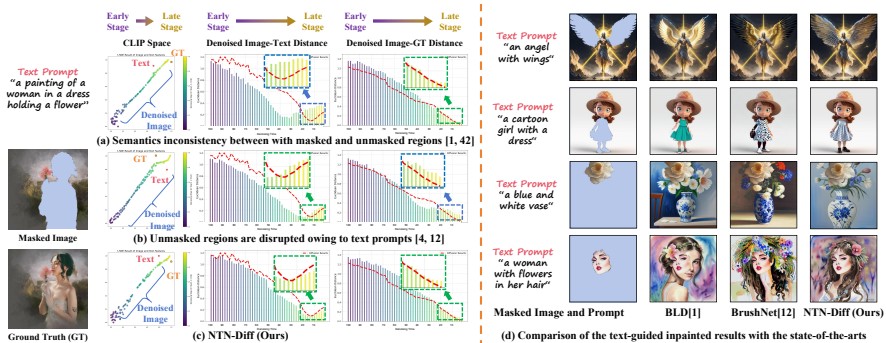

Figure 1: (a-c) t-SNE visualization of CLIP latent space evolution during text-guided denoising. At each step, the Euclidean distance between the text prompt and the denoised image (Denoised Image-Text Distance) reflects semantic consistency across masked and unmasked regions—a smaller value indicates better alignment. As a reference, we include the distance between the ground truth and the text prompt (red dashed line). To assess unmasked region preservation, we also compute the distance between the denoised image and ground truth (Denoised Image-GT Distance). (d) Comparison between our NTN-Diff (Fig.3) and state-of-the-arts [1, 12] for text-guided inpainting;

subtask in the broader field of image editing [20, 8, 7, 11, 15, 22], text-guided image inpainting distinguishes itself from other subtasks by requiring the masked regions to be generated as per text prompt, while preserve the unmasked regions. Upon the alignment between the generated content for masked regions and the text prompts, the longstanding challenges lie in *the preservation for unmasked regions, while achieving the semantics consistency between unmasked and masked regions as inpainted.*

To circumvent the above two challenges, a series of text-guided diffusion models [42, 25, 1, 35] have been proposed for image inpainting. Specifically, BLD [1] implements a blending operation during the denoising process at each step to preserve the unmasked regions, while Smartbrush [42] adopts a multi-task based denoising process conditioned on both text and shape for each step, where the denoised unmasked region is substituted by unmasked regions from ground-truth. Despite the unmasked preservation, as illustrated in Fig.1(a), they still suffer from *the semantics inconsistency between with masked and unmasked regions*, owing to the discrepancy from the *diffusion* process and the inpainted maksed regions from the *text-guided denoising* process.

Orthogonally, great efforts have been spent [31, 47, 4, 17, 26, 49] on preserving the unmasked regions while ensuring the semantics consistency between masked and unmasked regions. For instance, CAT [4] utilizes the text prompts and unmasked regions in the CLIP latent space to form latent code-level semantics consistency constraints for image inpainting, which, however, suffer from the information loss for unmasked regions. BrushNet [12] generates dense texture feature map during the denoising process with no cross-attention in UNet [34], to inpaint masked regions by unmasked information for consistency. However, as illustrated in Fig.1(b), *the unmasked regions fail to be preserved*, which is incurred by the other text-guided denoising process to inpaint masked regions.

So far, the prior arts still fail to *simultaneously* overcome the above two challenges. Such failure, as investigated in Fig.2, stems from the following hurdle: as disclosed by [29], the low-frequency band fluctuates more significantly during the early stage of the denoising process with high-level noise than the mid-and-high frequency bands, since diffusion models typically recover the low-frequency band first and progressively refine the mid-and-high frequency bands[1] (***See more results in Sec.A of the Appendix***). Therefore, the *low-frequency* band for both masked and unmasked regions are easy to be modulated by text prompts, as illustrated in Fig.2(a). To be contrary, as seen in Fig.2(b), the *mid-frequency* band across all regions is robust to the text prompts while aligns well with text prompts, which may better preserve the unmasked regions than low-frequency band upon text prompts. The above observations further motivate us to decompose the semantics consistency across masked and unmasked regions into the individual task of achieving that as per each frequency band across masked

---

[1]Compared to the mid-frequency band, the high-frequency band is sparser and more easy to be disrupted by high-level noise in the early stages, resulting in much less contribution on guiding the low-frequency band than mid-frequency band. Therefore, we focus on mid-and-low frequency bands.

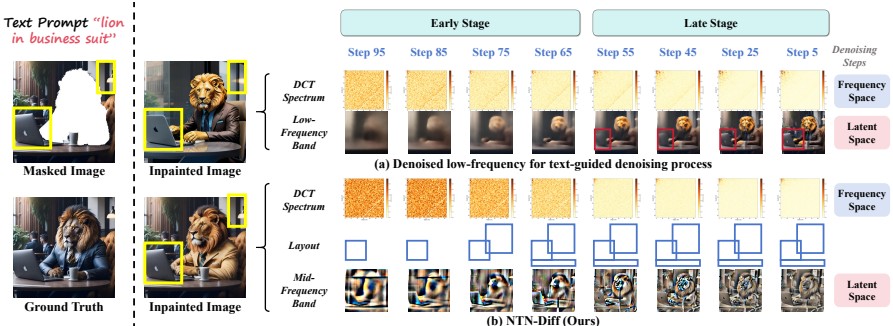

Figure 2: We investigate the text-guided denoising process for both the (a) low-frequency and (b) mid-frequency bands. For each step, we employ the red bounding boxes to highlight the variations for the low-frequency band during the late stage in (a) and the blue bounding boxes to visualize the variations for layout information in (b). For DCT spectrum, the top-left region represents low frequencies, with the bottom-right region corresponds to high frequencies. The dark red and yellow indicate the highest and lowest value.

and unmasked regions, while preserving the unmasked regions, to tackle two challenges in a row. To this end, we formally delve into the following questions: How to disentangle different frequency bands, particularly the *early* stage of the denoising process with high-level noise? and How to exploit the hybrid frequency bands for diffusion models to simultaneously achieve the above two goals?

Inspired by the recent progress on frequency-domain based diffusion models [39, 14, 10], we answer the above questions by proposing a frequency-aware null-text-null diffusion models, named NTN-Diff (see Fig.3), which comprises *early* and *late* stages for text-guided image inpainting. For the *early* stage, 1) we first propose a *null-text denoising process* (Sec.2.3.1) to avoid the low-frequency band is influenced by text prompts under the high-level noise, then replace its denoised result with unmasked regions from the forward diffused results. In parallel, 2) we perform the second *text-guided denoising process* (Sec.2.3.2) to denoise the masked regions, especially its mid-frequency band to be aligned with text prompts across both masked and unmasked regions, while replace the low-frequency band from the denoised output with that from the above null-text denoising process, so that the low-frequency can be preserved without being disturbed by text prompts; 3) we further exploit the above denoised mid-frequency to guide the last *null-text denoising process* (Sec.2.3.3), by substituting mid-frequency band from this null-text denoising process, while denoising the low-frequency band especially for masked regions, to attempt the complete alignment with text prompts.

Based on the above, during the *late* stage, we perform the *text-guided denoising process* (Sec.2.4), together with the unmasked regions substitution from the *early* stage of the diffusion process for unmasked regions preservations for each step, while achieve the semantics consistency between mid-and-low frequency bands across both masked and unmasked regions conditioned on text prompts, to well resolve the two challenges, leading to final inpainted output in Fig.1(c). Fig.1(d) further validates the intuition of NTN-Diff, which not only can achieve better semantics consistency between masked and unmasked regions than BLD [1], but also preserves the unmasked regions to outperform BrushNet [12], upon the alignment with text prompts.

The above observations further validate the intuitions of NTN-Diff by resolving the above two challenges, *i.e.*, semantics consistency between masked and unmasked regions and unmasked regions preservation, for the text-guided image inpainting via diffusion models. The extensive experimental results validate its superiority over the state-of-the-arts for text-guided image inpainting.

## 2 Methodology

Central to our proposed NTN-Diff lies in the following aspects: 1) the motivation of hybrid frequency aware diffusion models to text-guided image inpainting (Sec.2.2); 2) how to disentangle different frequency bands and exploit the hybrid frequency bands for diffusion models in the early stage (Sec.2.3); 3) how to achieve unmasked regions preservation and semantics consistency with masked regions in the late stage (Sec.2.4); before shedding light on NTN-Diff, we elaborate the preliminaries regarding DDPMs, which plays a pivotal role of text-guided image inpainting.

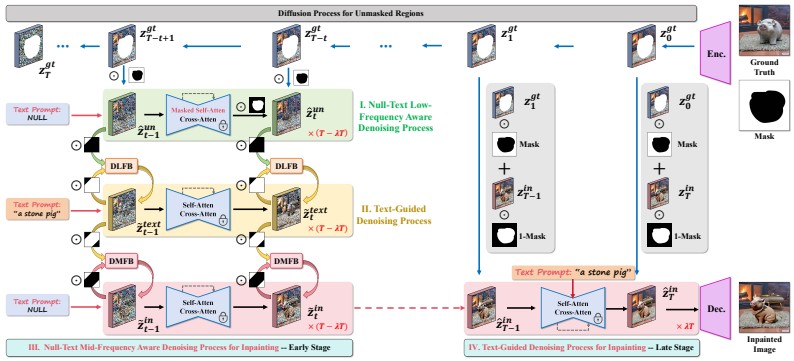

Figure 3: Illustration of our proposed NTN-Diff pipeline, which comprises a (I) *null-text denoising process* (Sec.2.3.1) to avoid being influenced by text prompts, and a (II) *text-guided denoising process* (Sec.2.3.2) to denoise the masked regions, while replacing the low-frequency band from the denoised output with that from the above null-text denoising process. Building on this, we further utilize the denoised mid-frequency to guide another (III) *null-text denoising process* (Sec.2.3.3) by substituting the mid-frequency band from this process. Additionally, a (IV) late-stage text-guided denoising process (Sec.2.4) is performed, along with the substitution of unmasked regions from the early stage of the diffusion process, to preserve unmasked regions at each step.

## 2.1 Preliminaries

Given a completed image $I_{gt} \in \mathbb{R}^{3 \times H \times W}$, a text prompt $c$, and a binary mask $M \in \{0, 1\}^{1 \times H \times W}$ such that 0 denotes masked regions and 1 denotes unmasked regions, The goal of text-guided image inpainting is to restore the missing regions in a masked image $I_m = I_{gt} \odot M$, so that the reconstructed content aligns with the text prompts to precisely indicate *where* and *what* to be inpainted. To this end, state-of-the-arts predominantly leverage pre-trained text-to-image latent diffusion frameworks, mainly for Stable Diffusion [33] (SD) as their foundation, where Variational Auto-Encoder (VAE) [13] is employed via encoder $\mathcal{E}$ to transform the image $I_{gt}$ from pixel space to latent space $z^{gt} \in \mathbb{R}^{4 \times \frac{H}{8} \times \frac{W}{8}}$, to reduce computational complexity without compromising visual quality. The model then performs the forward diffusion process and denoising process in the latent space:

**Forward Diffusion Process.** In the forward diffusion process with $T$ timesteps, following the typical Denoising Diffusion Probabilistic Model (DDPM) [9], Stable Diffusion adds Gaussian noise $\epsilon \sim \mathcal{N}(0, I)$ to convert the clean sample $z_0 = z^{gt}$ into a noisy sample $z_T$. For any timestep $t \in [0, T]$, the forward process $z_t$ is then defined as:

$$z_t(z_0, \epsilon) = \sqrt{\bar{\alpha}_t} z_0 + \sqrt{1 - \bar{\alpha}_t} \epsilon, \tag{1}$$

where $\bar{\alpha}_t$ denotes the corresponding noise level.

**Reverse Denoising Process.** During the reverse denoising process, Stable Diffusion is trained to estimate the noise added to the noisy image, conditioned on $c$, and progressively removes it over $T$ timesteps. Given an input noise $z_T$ sampled from a random Gaussian distribution, the training objective of the denoising network (UNet [34]) $\epsilon_\theta$ at the $t$-th timestep is formulated as:

$$\mathcal{L}_{LDM} = \mathbb{E}_{z_0, c, t \sim \mathcal{U}(0, T), \epsilon \sim \mathcal{N}(0, I)} \| \epsilon_\theta(z_t, t, \tau_\theta(c)) - \epsilon \|_2^2, \tag{2}$$

where $\tau_\theta(\cdot)$ is the CLIP [30] text encoder, and $\| \cdot \|_2$ denotes the $\ell_2$ norm. After $T$ timesteps, the denoised latent representation $z_0'$ is reconstructed into pixel space via the decoder $\mathcal{D}$ to generate the final result $I_{gen}$.

## 2.2 Motivation: Hybrid Frequency Aware Diffusion Models to Text-Guided Image Inpainting

Before shedding light on our pipeline, upon the diffusion process as per Eq.(1), we investigate the text-guided denoising process for both the denoised *low-frequency* (dense texture [17]) and *mid-frequency* (extracted from the Discrete Cosine Transform spectrum via mid-pass mask [5, 6]) information. The compared results are shown in Fig.2, while observe the followings:

(1) As shown in Fig.2(a), the low-frequency band can be easily changed by text prompts, especially the early stage with high-level noise, such trend keeps on even into nearly the middle (*e.g.,* 45-th step) of the late stage.

(2) Fig.2(b) illustrates the text-guided denoising results for mid-frequency band. To validate its stability, we visualize the layout information as bounding box for each step (2nd row), which, as indicated by [5, 6, 43], is closely related to the mid-frequency band. Akin to Fig.2(a), the denoised mid-frequency band (3rd row) also changes during the initial early stage owing to text prompts with high-level noise, yet quickly converges by the end (*e.g.,* 65-th step) of the early stage, which further leads to the stable mid-frequency band with *nearly* no influence by text prompt across the whole late stage with low-level noise.

Capitalizing on the discoveries, instead of low-frequency, we propose to exploit the *mid-frequency* band during denoising process, which plays the pivotal role of achieving the semantics consistency upon text prompts, while can better achieve the semantics consistency between masked and unmasked regions, while preserve its own frequency band well, than low-frequency band, owing to its *robustness* to the text prompt during the denoising process. The above observations further motivate us with a novel frequency-aware null-text-null diffusion models, named NTN-Diff, for text-guided image inpainting. Our pipeline is illustrated in Fig.3, where we discuss NTN-Diff within the *early* and *late* stages, demarcated by the critical step $\lambda T$ with T as the total steps for diffusion models; the detailed discussions for $\lambda$ can be seen in Sec.3.3.2. (***See more intuitions on hybrid frequency aware diffusion models to text-guided image inpainting in Sec.B of the Appendix***)

### 2.3 Early Stage for Null-Text-Null Frequency-Aware Diffusion Models

Based on the diffusion process regarding Eq.(1), we discuss the early stage of text-guided denoising process. Unlike the typical models [42, 1] with text-guided denoising process, 1) we propose a *null-text denoising process* (Sec.2.3.1) conditioned on null-text prompt at the $t$-th step to avoid being influenced by text prompts even under the high-level noise, focusing primarily on low-frequency band, we then replace its denoised result with the unmasked regions from the forward diffused results within the $(T - t)$-th step. In parallel, 2) we perform the second *text-guided denoising process* (Sec.2.3.2) for $t$-th step, to denoise the masked regions, while replace the low-frequency band (Fig.4(a)) from the denoised output with that from the above null-text denoising process, so that low-frequency band (*e.g.*, color and illumination) especially for unmasked regions can be preserved even under text prompts, while the mid-frequency band over both masked and unmasked regions varies owing to text prompts, yet struggling to be semantically consistent to its low-frequency band across the whole regions, resulting in semantic inconsistency between masked and unmasked regions (see Fig.6).

The denoised mid-frequency band well aligns with the text prompts especially for masked regions, while also encodes the information related to low-frequency band from the above null-text denoising process (Sec.2.3.3), which is further exploited to guide 3) the last *null-text denoising process*, to 3.1) achieve semantics consistency between mid-and-low frequency bands across masked and unmasked regions, by denoising the low-frequency band throughout the path to be semantically consistent to mid-frequency band, with no influence from text prompts (see Fig.2(a)); 3.2) such intuition can further restore low-frequency band especially for masked regions, to be aligned with text prompts, while ready to preserve both mid-and-low frequency bands for unmasked regions, along with its semantics consistency to masked regions via a late stage *text-guided denoising process* in Sec.2.4.

#### 2.3.1 Null-Text Low-Frequency Aware Denoising Process

Formally, we initialize the denoising process with random Gaussian noise $z_T^{un} \in \mathbb{R}^{4 \times \frac{H}{8} \times \frac{W}{8}}$ and null-text $c_\emptyset$ as the input to the denoising network $\epsilon_\theta(z_T^{un}, T, \tau_\theta(c_\emptyset))$ for *null-text denoising process*. To preserve the unmasked regions, at each step $t$, we replace the unmasked regions of denoising result $z_t^{un}$ with those from the diffusion result $z_{T-t}^{gt}$ at the $(T - t)$-th step to yield $\hat{z}_t^{un}$, as per the following:

$$\hat{z}_t^{un} = z_{T-t}^{gt} \odot m_z + z_t^{un} \odot (1 - m_z), \tag{3}$$

where $m_z \in \mathbb{R}^{1 \times \frac{H}{8} \times \frac{W}{8}}$ is obtained by downsampling the mask $M$, we replicate it along four channels for $z_{T-t}^{gt}$ and $z_t^{un}$, $\odot$ is the element-wise multiplication. Following the above, we reformulate the self-attention mechanism of the denoising network by incorporating a mask to suppress the attention scores of the masked regions, to ensure that the inpainting process primarily relies on the unmasked regions. Given the denoised latent image feature $F_l \in \mathbb{R}^{(h_l \times w_l) \times d_l}$ at the $l$-th layer of denoising UNet architecture $\epsilon_\theta(z_t^{un}, t, \tau_\theta(c_\emptyset))$ at the $t$-th timesteps, the modified self-attention mechanism is

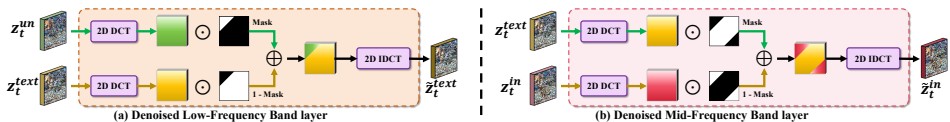

Figure 4: Illustration of (a) denoised low-frequency band layer and (b) mid-frequency band layer.

defined as follows:

$$F_{l+1} = \text{softmax}\left(\frac{\left(F_l W_{Q_l^s}\right)\left(F_l \odot m_l W_{K_l^s}\right)^T}{\sqrt{d_l}}\right) F_l W_{V_l^s}, \tag{4}$$

where $m_l \in \mathbb{R}^{h_l \times w_l}$ is the downsampled result of the mask $M$, we replicate it along $d_l$ channels for $F_l$, $W_{Q_l^s} \in \mathbb{R}^{d_l \times d_{l+1}}$, $W_{K_l^s} \in \mathbb{R}^{d_l \times d_{l+1}}$ and $W_{V_l^s} \in \mathbb{R}^{d_l \times d_{l+1}}$ are learnable linear projection matrices, with $d_l$ representing the embedding dimensions.

### 2.3.2 Text-Guided Denoising Process

To inpaint the masked regions as per text prompts, we resort to the *text-guided denoising process*, together with low-frequency band substitution from null-text denoising process (Sec.2.3.1). Specifically, we initialize the denoising process with random Gaussian noise $z_T^{text} \in \mathbb{R}^{4 \times \frac{H}{8} \times \frac{W}{8}}$ and text prompts $c$ as the input to the denoising network $\epsilon_\theta(z_T^{text}, T, \tau_\theta(c))$ for text-guided denoising process for masked regions. As inspired by Fig.2, we preserve the unmasked regions as per their low-frequency band, since, instead of mid-frequency or even high-frequency band, such information can be easily modulated by text prompt. To this end, we propose a plug-and-play Denoised Low-Frequency Band layer (Fig.4(a)) to substitute the latent representation $z_t^{text}$ within the text-guided denoising process, by $z_t^{un}$ from the null-text denoising process (see Sec.2.3.1) for the $t$-th step to yield $\tilde{z}_t^{text}$, which is formulated as:

$$\tilde{z}_t^{text} = \text{IDCT}\left(\text{DCT}(z_t^{un}) \odot m_{low} + \text{DCT}(z_t^{text}) \odot (1 - m_{low})\right), \tag{5}$$

where $\text{DCT}(\cdot)$ represents 2D Discrete Cosine Transform, $\text{IDCT}(\cdot)$ represents 2D Inverse DCT, and $m_{low}$ denotes the low-pass mask to isolate low-frequency band, the denoised low-frequency band layer employs a binary mask defined by the sum of the 2D coordinates as a threshold, as follows:

$$m_{low}(x, y) = \begin{cases} 1, & \text{if } x + y \leq th_{lp} \\ 0, & \text{otherwise} \end{cases} \tag{6}$$

where $th_{lp}$ is the threshold for low-pass filtering. To adaptively extract the low-frequency band from masked images, we set $th_{lp}$ as follows:

$$th_{lp} = \lambda_{lp}^f + \lambda_{lp}^r \cdot \frac{\|M\|_1}{HW}, \tag{7}$$

where $\lambda_{lp}^f$ and $\lambda_{lp}^r$ are hyper-parameters. The $\ell_1$ norm $\|M\|_1$ represents the number of entry with value 1 in $M$. We set $th_{lp}$ to correlate with the ratio of unmasked regions, since the larger unmasked regions require more low-frequency band from the null-text denoising process to substitute the low-frequency band within the text-guided denoising process.

**Remark 1.** During this process, the low-frequency band across both masked and unmasked regions are preserved even under text prompts owing to the substitutions from the null-text denoising process, with only mid-frequency aligned with text prompts for both regions, hence still failed to achieve the consistency between masked and unmasked regions. To address this, we propose another concurrent *null-text denoising process*, guided by above denoised mid-level frequency, to help denoise the low-frequency band for both regions, especially for masked regions.

### 2.3.3 Null-Text Mid-Frequency Aware Denoising Process

We resort to another *null-text denoising process* guided by the denoised mid-frequency band from the second text-guided denoising process (Sec.2.3.2) for the $t$-th step of the early stage. Particularly, we initialize the denoising process with random Gaussian noise $z_T^{in} \in \mathbb{R}^{4 \times \frac{H}{8} \times \frac{W}{8}}$ and null-text $c_\emptyset$ as input to the denoising network $\epsilon_\theta(z_T^{in}, T, \tau_\theta(c_\emptyset))$. We further exploit the denoised mid-frequency result from $z_t^{text}$, to substitute the denoised mid-frequency band of $z_t^{in}$ from the last *null-text denoising*

*process*, throughout a plug-and-play Denoised Mid-Frequency Band layer (as illustrated in Fig.4(b)) to yield $\tilde{z}_t^{in}$, which is formulated below:

$$\tilde{z}_t^{in} = \text{IDCT}\left(\text{DCT}(z_t^{text}) \odot m_{\text{mid}} + \text{DCT}(z_t^{in}) \odot (1 - m_{\text{mid}})\right), \tag{8}$$

where $m_{\text{mid}}$ denotes the mid-pass mask. Analogous to Eq. (6), we define:

$$m_{\text{mid}}(x, y) = \begin{cases} 1, & \text{if } th_{mp1} < x + y \leq th_{mp2} \\ 0, & \text{otherwise} \end{cases} \tag{9}$$

where $th_{mp1}$ and $th_{mp2}$ are the thresholds for mid-pass filtering to extract mid-frequency band. To adaptively extract the mid-frequency band from masked images, we set $th_{mp1}$ and $th_{mp2}$ as follows:

$$th_{mp1} = \lambda_{mp1}^f - \lambda_{mp1}^r \cdot \frac{\|M\|_1}{HW}, th_{mp2} = \lambda_{mp2}^f + \lambda_{mp2}^r \cdot \frac{\|M\|_1}{HW}, \tag{10}$$

where $\lambda_{mp1}^f$, $\lambda_{mp1}^r$, $\lambda_{mp2}^f$ and $\lambda_{mp2}^r$ are hyper-parameters, such that $th_{mp2} > th_{mp1}$. Similar as Eq. (7), we also set the $th_{mp1}$ and $th_{mp2}$ to correlate with the ratio of unmasked regions. Since the larger unmasked regions require more mid-frequency band to encode the information from the low-frequency band substituted from the first null-text denoising process.

**Remark 2.** The above null-text mid-frequency guided denoised process can denoise the low-frequency band for the masked regions to be aligned with text prompt, owing to *the denoised mid-frequency from text-guided process*; besides, the denoised mid-and-low frequency bands from unmasked regions also help denoising process for masked regions, which, however, cannot be served as the final text-guided inpainting output for masked regions, as the denoised mid-and-low frequency band is not semantically strong as text prompt. To address that, a *text-guided denoising process* is performed as the late stage of denoising process, as discussed below.

## 2.4 Late Stage of Text-Guided Denoising Process

Following the discussion above, for final late stage of denoising process, we perform the *text-guided denoising process* based on the early null-text mid-frequency guided denoising process (Sec.2.3.3), the low-frequency band within the masked regions is denoised throughout the path to text prompts alignment, together with the stable mid-frequency band (see Fig.2(b)) from the early text-guided denoising process (Sec.2.3.2), the masked regions are well inpainted. However, the low-frequency band for unmasked regions biases a lot from the ground truth from the early null-text denoising process (Sec.2.3.3), and augmented by the influence from the text prompts for late stage. To this end, we substitute the unmasked regions from early stage of the diffusion process into the denoising results $z_t^{in}$ for the $t$-th step during the late stage to yield $\hat{z}_t^{in}$, which is formulated as:

$$\hat{z}_t^{in} = z_{T-t}^{gt} \odot m_z + z_t^{in} \odot (1 - m_z), \tag{11}$$

where $z_{T-t}^{gt}$ donates the forward diffused results with the $(T - t)$-th step. Building upon the noisy sample $\hat{z}_t^{in}$, the text prompt $c$, along with the step $t$, we perform the denoising network $\epsilon_\theta(\hat{z}_t^{in}, t, \tau_\theta(c))$ to get the denoising result $z_{t-1}^{in}$ in the latent space for the next $(t - 1)$-th step. After $\lambda T$ steps, the denoised latent representation $z_0^{in}$ is reconstructed into the pixel space via the VAE decoder $\mathcal{D}$ to generate the final image inpainting output. (***See the pseudo algorithm in Sec.C of the Appendix***)

**Remark 3 : Last question on Semantics Consistency is answered!** Although the alignment between mid-and-low frequency bands from masked regions and text prompts, together with un-masked regions preservation, can be achieved, one last question is *how the denoised results within this late stage can achieve semantics consistency between masked and unmasked regions?* The crucial intuition lies in the denoised mid-frequency band from the early text-guided denoising process (Sec.2.3.2), which, based on **Remark 2**, also encodes the information from the low-frequency band substituted from the null-text denoising process to match diffusion process (ground truth) including both masked and unmasked regions, hence the low-frequency band under mid-frequency guidance for masked regions also achieves the consistency to the substituted unmasked regions from the diffusion process, since the mid-frequency band is *stable* during the late stage of the denosing process *even under text prompts* (see Fig.2(b)).

With all frequency bands to be semantically consistent across all regions conditioned on text prompts, the *semantics consistency* between masked and unmasked are well achieved, together with the success of the *preservation for unmasked regions*!

Table 1: Quantitative comparisons between NTN-Diff (*) and other diffusion-based inpainting models over BrushBench for inpainting and outpainting are shown, where all models use **Stable Diffusion V1.5** as the baseline model. ↑: higher is better; ↓: lower is better; red and blue stand for the best and second best result. * with blending operation of BrushNet. NTN-Diff (*) achieves the best result.

| Task | Models | Venue | IR$_{\times 10}$↑ | HPS v2$_{\times 10^2}$↑ | PSNR↑ | MSE$_{\times 10^3}$↓ | LPIPS$_{\times 10^3}$↓ | CLIP Score↑ |
|---|---|---|---|---|---|---|---|---|
| **Inside Inpainting** | **BLD**[1] | TOG' 23 | 9.78 | 25.87 | 21.33 | 9.76 | 49.26 | 26.15 |
| | **CNI**[47] | ICCV' 23 | 9.9 | 26.02 | 12.39 | 78.78 | 243.62 | 26.47 |
| | **PP**[49] | ECCV' 24 | 11.46 | 27.35 | 21.43 | 32.73 | 48.43 | 26.48 |
| | **BrushNet**[12] | ECCV' 24 | 12.36 | 27.40 | 21.65 | 9.31 | 48.28 | 26.48 |
| | **HDP**[25] | ICLR' 25 | 11.68 | 26.90 | 22.61 | 9.95 | 43.50 | 26.37 |
| | **NTN-Diff (Ours)** | - | 12.45 | 27.57 | 23.51 | 6.50 | 40.79 | 26.54 |
| | **CNI***[47] | ICCV' 23 | 11.21 | 26.92 | 22.73 | 24.58 | 43.49 | 26.22 |
| | **BrushNet***[12] | ECCV' 24 | 12.64 | 27.78 | 31.94 | 0.80 | 18.67 | 26.39 |
| | **NTN-Diff* (Ours)** | - | 12.69 | 27.82 | 40.70 | 0.11 | 0.88 | 26.49 |
| **Outside Inpainting** | **BLD**[1] | TOG' 23 | 7.81 | 26.77 | 15.85 | 35.86 | 21.40 | 26.73 |
| | **CNI**[47] | ICCV' 23 | 9.26 | 27.68 | 11.91 | 83.03 | 58.16 | 27.29 |
| | **PP**[49] | ECCV' 24 | 7.45 | 28.01 | 18.04 | 31.78 | 15.13 | 26.72 |
| | **BrushNet**[12] | ECCV' 24 | 10.82 | 28.02 | 18.06 | 22.86 | 15.08 | 27.33 |
| | **HDP**[25] | ICLR' 25 | 9.66 | 27.79 | 18.03 | 22.99 | 15.22 | 26.96 |
| | **NTN-Diff (Ours)** | - | 11.54 | 28.22 | 18.47 | 20.44 | 14.46 | 27.54 |
| | **CNI***[47] | ICCV' 23 | 9.57 | 27.76 | 17.50 | 37.72 | 19.95 | 26.92 |
| | **BrushNet***[12] | ECCV' 24 | 10.88 | 28.09 | 27.82 | 2.25 | 4.63 | 27.22 |
| | **NTN-Diff* (Ours)** | - | 11.61 | 28.36 | 31.08 | 1.23 | 1.24 | 27.30 |

# 3 Experiments

## 3.1 Implementation Details

We evaluate NTN-Diff on two benchmarks tailored for text-guided image inpainting in diffusion models: **BrushBench** [12] comprises 600 images with human-annotated masks and captions. The dataset balances natural and artificial images (*e.g.*, paintings) and equally distributes across categories (humans, animals, indoor/outdoor scenarios), ensuring fair and equitable evaluation, BrushBench refines the task by considering two specific scenarios: segmentation mask inside-inpainting and segmentation mask outside-inpainting; **EditBench** [35] features 240 images with an equal split between natural and generated images, each accompanied by mask and caption annotations. Following [12], we utilize a fine-tuned version of **Stable Diffusion v1.5** as the base model, configured with 100 sampling steps to ensure high-quality outputs. All experiments are implemented in PyTorch and run on a single NVIDIA RTX 3090.

We evaluate inpainted results using six metrics across three aspects. For ***image generation quality***, we employ **Image Reward (IR)** [44] to model human preferences for image quality and **HPS V2** [40] that combines perceptual studies with deep learning for comprehensive visual and semantic assessment. Both metrics measure the *semantics consistency* of the high-quality image generation aligned with human standards; for ***masked region preservation***, we use **PSNR** and **MSE** to measure pixel-wise differences from ground truth, along with **LPIPS** [48] which evaluates perceptual similarity through deep feature distances; finally, for ***text alignment***, **CLIP Score** [38] quantifies *text-image consistency* by comparing their embeddings in CLIP's shared space.

## 3.2 Comparison with State-of-the-arts

To validate the superiority of **NTN-Diff**, we conduct a comprehensive comparison with state-of-the-art text-guided image inpainting diffusion models, including **HDP** [25] and **BLD** [1] which implement a blending operation during the denoising process at each step to preserve the unmasked regions, while suffer from the semantics inconsistency between with masked and unmasked regions, owing to the discrepancy between the diffusion process for unmasked regions substitution and the denoising process for masked region alignment with text prompt. **CNI** [47], **PP** [49] and **BrushNet** [12] focus on the ensuring the semantics consistency between masked and unmasked regions. Particularly, **BrushNet** [12] generates dense texture feature map during the denoising process with no cross-attention in UNet [34], to inpaint masked regions by unmasked information for consistency, while unmasked regions are disrupted owing to another text-guided denoising process.

**Quantitative and Qualitative analysis.** The quantitative results in Table.1 summarize our findings below: **NTN-Diff** enjoys larger IR, HPS v2, PSNR and CLIP Score, together with smaller MSE and LPIPS than the competitors. Notably, **BrushNet** remains the large performance margins (at most 0.73% for IR, 0.62% for HPS, 8.59% for PSNR, 30.18% for MSE, 15.51% for LPIPS and 0.23% for CLIP Score) compared to **NTN-Diff** in Table.1. We also present the quantitative results of **NTN-Diff** with the pixel-level blending operation of [12], named **NTN-Diff\***, to preserve the unmasked regions,

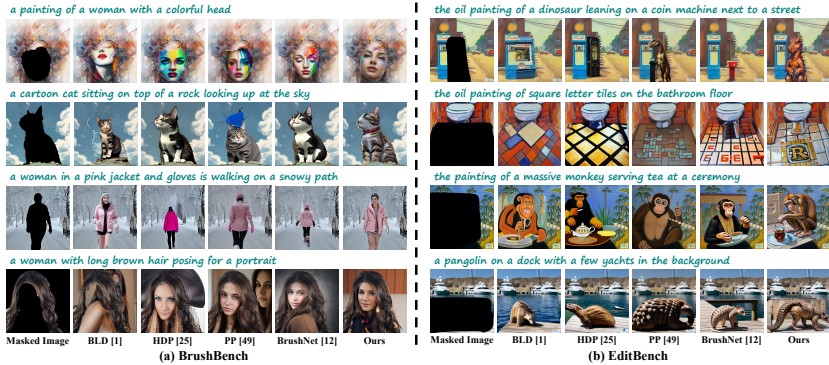

Figure 5: Comparison of the text-guided inpainted results with the state-of-the-arts on BrushBench [12] and EditBench [35]. NTN-Diff delivers the superior inpainted results over others.

Table 2: Ablation studies on three denoising processes: Case A (Sec.2.3.1), Case B (Sec.2.3.2) and Case C (Sec.2.3.3) for the early stage, based on the same text-guided denoising process (Sec.2.4) for the late stage. **red** and **blue** stand for the best and second best result.

Table 3: Ablation study about the hyperparameter sensitivity analysis of $\lambda$. **red** and **blue** stand for the best and second best result. With $\lambda = 0.6$ balancing early and late stage, the best performance of image quality, unmasked region preservation and text alignment are achieved.

| Metric | EditBench | | | | BrushBench | | | |
|---|---|---|---|---|---|---|---|---|
| | Case A | Case B | Case C | Ours | Case A | Case B | Case C | Ours |
| $IR_{\times 10}\uparrow$ | 2.41 | 1.24 | 2.14 | 3.10 | 10.14 | 9.59 | 10.02 | 11.12 |
| PSNR↑ | 22.36 | 22.39 | 22.53 | 22.65 | 28.02 | 27.71 | 28.06 | 28.10 |
| $LPIPS_{\times 10^3}\downarrow$ | 26.60 | 24.82 | 25.24 | 24.21 | 44.54 | 47.08 | 44.92 | 44.09 |
| CLIP Score↑ | 28.62 | 28.53 | 28.60 | 28.95 | 25.95 | 25.78 | 26.03 | 26.09 |

| Metric | EditBench | | | | | BrushBench | | | | |
|---|---|---|---|---|---|---|---|---|---|---|
| | 0.9 | 0.8 | 0.7 | 0.6 | 0.5 | 0.9 | 0.8 | 0.7 | 0.6 | 0.5 |
| $IR_{\times 10}\uparrow$ | 2.41 | 2.67 | 2.78 | 3.10 | 1.10 | 10.56 | 10.40 | 10.76 | 11.12 | 10.77 |
| PSNR↑ | 22.03 | 22.19 | 22.44 | 22.65 | 22.63 | 27.69 | 27.86 | 28.04 | 28.10 | 28.08 |
| $LPIPS_{\times 10^3}\downarrow$ | 30.10 | 29.28 | 26.95 | 24.21 | 23.16 | 49.26 | 46.85 | 44.88 | 44.09 | 43.44 |
| CLIP Score↑ | 28.89 | 28.76 | 29.20 | 28.95 | 28.81 | 25.87 | 26.01 | 25.93 | 26.09 | 25.91 |

which demonstrates the ability to tame the hybrid frequency issue, the results further verifies the intuition in Sec.1 – NTN-Diff achieves the *semantics consistency* between mid-and-low frequency bands across masked and unmasked regions, while *preserving unmasked regions*. We illustrate the above intuitions in Fig.5. (***See more analysis of computational efficiency in Sec.D of the Appendix, along with more compared results for Editbench [35] can be seen in Sec.E of the Appendix***)

## 3.3 Ablation Studies

### 3.3.1 Discussion on Different Denoising Processes of the Early Stage

To validate of our NTN-Diff (Sec.2.3), we perform the ablation study on BrushBench and EditBench datasets with three variants: **Case A**: removing masked self-attention in the first null-text denoising process (Sec.2.3.1); **Case B**: removing both null-text and text-guided denoising process (Sec.2.3.2). **Case C**: removing the null-text denoising process (Sec.2.3.3); Table.2 reports the results for all three cases, which suggests that NTN-Diff outperforms **Case A**, confirming that *the null-text denoising process conditioned on null-text prompt can avoid being influenced by text prompts even under the high-level noise, focusing primarily on low-frequency band,* which in line with Sec.2.3.1; **Case B** exhibits a suboptimal performance degradation, despite the null-text denoising process in the early stage, together with the text-guided denoising process, which verifies that the null-text mid-frequency guided denoised process can denoise the low-frequency band for the masked regions to be aligned with text prompt, owing to *the denoised mid-frequency from text-guided process*, hence validating the important of text-guided process (Sec.2.3.2); NTN-Diff maintains achieves the large performance gain over **Case C** (at most 44.8%), implying that only mid-frequency *aligned with text prompts for masked and unmasked regions*, still failed to achieve the consistency across the whole regions, which is in line with Sec.2.3.3 and illustrated in Fig.6. (***See more generation results for the impact of the text and null-text prompts on low-and-mid frequency bands in Sec.F of the Appendix***).

### 3.3.2 Hyperparameter Sensitivity Analysis of $\lambda$ for the Length of the Early and Late Stage

We analyze the parameter $\lambda$ (Sec.2.2) that divides the denoising process into *early* and *late* stages at step $\lambda T$, we set the $\lambda$ values to {0.9; 0.8; 0.7; 0.6; 0.5}. As reported in Table.3, when $\lambda = 0.9$, the performance is the worst with the shortest early stage. When $\lambda = 0.6$ with the balance of early and late stage, the best performance are achieved, confirming that the desirable length of early stage

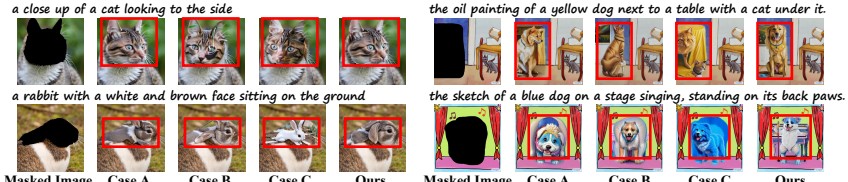

Figure 6: The inpainted output about the different denoising process for the early stage, NTN-Diff delivers the better inpainted results (marked as the red box) than others. **Case A** fails to preserve the unmasked region in the null-text denoising process (Sec.2.3.1), such as the background of dog (4th row); without text-guided denoising process (Sec.2.3.2), **Case B** cannot inpaint the masked region as per the text prompts, such as the disappeared dog in the 3rd row of Case B; specifically, **Case C** cannot achieve the semantics consistency between masked and unmasked region, owing to the null-text denoising process (Sec.2.3.3), such as the tiny rabbit within the rabbit head-shaped mask.

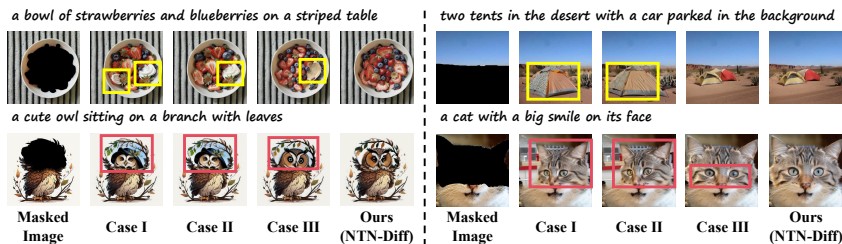

Figure 7: The inpainted output about the impact of adaptively extracting the low-and-mid frequency bands, NTN-Diff can achieve the better inpainted results (marked as the **box**) than others.

can make the *low-frequency* band under the *mid-frequency guidance* for masked regions *achieve the consistency* to the substituted unmasked regions for ground truth, which is consistent to Sec.2.4. (***See more generation results in Sec.G of the Appendix***).

### 3.3.3 More Ablation Studies on the Impact of Adaptively Extracting the Low-and-Mid Frequency Bands

As mentioned in Sec.2.3.2 and Sec.2.3.3, we set $th_{lp}$ in Eq.(5), and $th_{mp1}$ and $th_{mp2}$ in Eq.(8) to correlate with the ratio of unmasked regions. We provide ablation studies on the impact of adaptively extracting low-and-mid frequency bands on the BrushBench and EditBench datasets using three variants: **Case I** adopts fixed thresholds to extract both mid-and-low frequency bands for different images; **Case II** adaptively extracts low-frequency band while using fixed thresholds for mid-frequency bands; **Case III** adaptively extracts mid-frequency band while using a fixed threshold for low-frequency extraction. As shown in Fig.7, due to the use of fixed thresholds for extracting mid-frequency band, **Case I** and **Case II** inevitably generate inconsistent content between masked and unmasked regions (*e.g.*, the smiling cat in the second row of Fig. 7), which can be attributed to the fact *that the larger unmasked regions require more mid-frequency band to encode the information from the low-frequency band substituted from the first null-text denoising process*, as explained in Sec. 2.4. In contrast, **Case III** shows substantial performance degradation (e.g., the light yellow color blocks in the bowl in the first row of Fig.7), confirming *that larger unmasked regions demand more low-frequency information from the null-text denoising process to replace the low-frequency bands in the text-guided denoising process*, which is consistent with the findings in Sec.2.3.2.

## 4 Conclusion

In this paper, we propose a null-text-null frequency-aware diffusion models, named NTN-Diff, for text-guided image inpainting, by decomposing the semantics consistency across masked and unmasked regions into the consistencies as per each frequency band, while preserving the unmasked regions, to simultaneously address two challenges of unmasked regions preservations, along with its semantics consistency with inpainted masked regions. The extensive experimental results validate the advantages of NTN-Diff over state-of-the-art diffusion models for text-guided image inpainting.

**Acknowledgments** This research is supported by National Natural Science Foundation of China (U21A20470, 62172136, 72188101); Institute of Advanced Medicine and Frontier Technology (2023IHM01080), and sponsored by CCF-NetEase ThunderFire Innovation Research Funding (NO. CCF-Netease 202513); The computation is completed on the HPC Platform of Hefei University of Technology.

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

## Appendix Overview

Due to page limitation of the mainbody, as indicated by our submission, the supplementary material offers further discussion on the motivation of mid-frequency band and more visual results with higher resolution, which are summarized below:

- More analysis of the high-and-low frequency components in the denoising process, as mentioned in Sec.1 of the mainbody (Sec.A)
- More *intuitions* on hybrid frequency aware diffusion models to text-guided image inpainting, as mentioned in Sec.2.2 of the mainbody (Sec.B)
- The algorithm of NTN-Diff, as mentioned in Sec.2.4 of the mainbody (Sec.C)
- More analysis of computational efficiency for the comparison with state-of-the-arts, as mentioned in Sec.3.2 of the mainbody (Sec.D).
- Additional quantitative results and qualitative analysis with *higher resolution* for the comparison with state-of-the-arts, as mentioned in Sec.3.2 of the mainbody (Sec.E).
- More ablation studies on the impact of text and null-text prompts on low-and-mid frequency bands during denoising process, as mentioned in Sec.3.3.1 of the mainbody (Sec.F).
- Additional generation results for the ablation study about hyperparameter sensitivity analysis of $\lambda$ for the length of the early and late stage, as mentioned in Sec.3.3.2 of the mainbody (Sec.G).
- We list the limitations and broader impacts (Sec.H).

## A  More Analysis of the High-and-Low Frequency Components in the Denoising Process

Due to the page limitation, we further conduct experiments to verify the conclusion stated in **Sec. 1 of the mainbody**: *"the low-frequency band fluctuates more significantly during the early stage of the denoising process with high-level noise than the mid- and high-frequency bands."* This observation aligns with the common understanding that diffusion models first recover low-frequency components and progressively refine mid- and high-frequency details. Motivated by this, we experimentally validate the denoising process using several metrics to measure variations in the low-and-high frequency bands at different timesteps, as reported in Table. 4.

We adopt two groups of metrics to analyze the frequency behavior during the denoising process:

- **Low-frequency information:** We adopt *Mean Gray*, *Mean HSV-V*, and *Mean Luma*, which primarily capture the global brightness and luminance structure of the denoised image. These metrics evaluate how well the low-frequency bands are maintained throughout the denoising process.
- **High-frequency information:** We use *Average Gradient*, *Variance*, and *LBP Variance*, which are sensitive to edge sharpness, local contrast, and texture complexity, respectively. These metrics reflect the preservation and recovery of fine details during denoising.

The results show that low-frequency metrics (*Mean Gray*, *HSV-V*, *Luma*) remain nearly unchanged from the beginning, indicating early stabilization. In contrast, high-frequency metrics (*Avg Gradient*, *Variance*, *LBP Variance*) increase steadily in the later timesteps, **confirming that high-frequency details are refined gradually during denoising**, which is consistent with the statement in **Sec. 1 of the mainbody**.

## B  More Intuition on Hybrid Frequency Aware Diffusion Models to Text-Guided Image Inpainting

Due to page limitation, we offer more visual results of the mid-frequency band of the denoising process. Fig.8(a) illustrates the text-guided denoising results for mid-frequency band. To validate its stability, we visualize the layout information as bounding box for each step, which, as indicated

Table 4: Metrics measuring the low-and-high frequency band variations during the denoising process, confirming that diffusion models typically recover the low-frequency band first and progressively refine the mid-and-high frequency bands.

| Timesteps | Mean Gray | Mean HSV-V | Mean Luma | Avg Gradient | Variance | LBP Variance |
|---|---|---|---|---|---|---|
| 0 | 120.9 | 136.0 | 120.9 | 59.01 | 3955.0 | 4.45 |
| 5 | 120.8 | 134.6 | 120.8 | 75.07 | 4098.3 | 3.71 |
| 10 | 120.7 | 133.5 | 120.7 | 101.58 | 4378.9 | 3.25 |
| 15 | 120.7 | 133.7 | 120.7 | 133.26 | 4790.8 | 3.07 |
| 20 | 120.9 | 135.1 | 120.9 | 164.23 | 5274.8 | 3.07 |
| 25 | 121.2 | 137.3 | 121.2 | 190.49 | 5743.9 | 3.15 |
| 30 | 121.4 | 139.2 | 121.4 | 209.84 | 6099.5 | 3.25 |
| 35 | 121.1 | 140.0 | 121.1 | 222.03 | 6291.1 | 3.32 |
| 40 | 120.5 | 140.1 | 120.5 | 228.75 | 6363.9 | 3.35 |
| 45 | 119.9 | 139.8 | 119.9 | 232.19 | 6394.8 | 3.38 |
| 50 | 119.5 | 139.5 | 119.5 | 233.88 | 6419.5 | 3.40 |

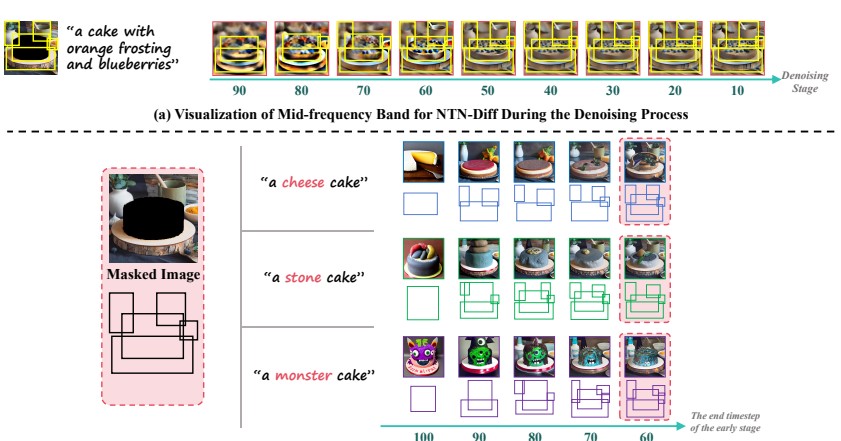

(a) Visualization of Mid-frequency Band for NTN-Diff During the Denoising Process

(b) More Results on Mid-frequency Band for NTN-Diff During the Denoising Process

Figure 8: We investigate the text-guided denoising process for the mid-frequency bands. For each step, we employ the bounding boxes to visualize the variations for layout information in (a) and the changes of layout information for the same image based on three similar text prompts. the denoised mid-frequency band changes during the initial early stage owing to text prompts with high-level noise, yet quickly converges by the end of the early stage, which further leads to the stable states with nearly no influence by text prompt across the whole late stage.

by [5, 6, 43], is closely related to the mid-frequency band. The denoised mid-frequency band also changes during the initial early stage owing to text prompts with high-level noise, yet quickly converges by the end of the early stage e.g., 60-th step, which further leads to the stable mid-frequency band with *nearly* no influence by text prompt across the whole late stage with low-level noise. We further exhibit additional visual results by performing the experiments on the same image based on three similar text prompts. Despite the different text prompts, the mid-frequency can quickly converge by the end of the early stage, *e.g.*, 60-th. Following the above, we propose to exploit the mid-frequency band during denoising process, which plays the pivotal role of achieving the semantics consistency upon text prompts, while can better achieve the semantics consistency between masked and unmasked regions, to further preserve its own frequency band well, owing to its robustness to the text prompt during the denoising process.

## C  The Algorithm of NTN-Diff

Our proposed NTN-Diff pipeline consists of a *null-text denoising process* (Sec. 2.3.1) that avoids the influence of text prompts, and a *text-guided denoising process* (Sec. 2.3.2) that reconstructs the masked regions while replacing the low-frequency band of its denoised output with that from the null-text denoising process. Building upon this design, we further utilize the denoised mid-frequency information to guide another *null-text denoising process* (Sec. 2.3.3) by substituting the mid-frequency

band accordingly. In addition, a late-stage *text-guided denoising process* (Sec. 2.4) is conducted, during which the unmasked regions from the early stage of the diffusion process are progressively substituted to preserve structural consistency. Based on the above, we summarize the complete algorithm of our proposed *null-text-null frequency-aware diffusion model* in Algorithm 1 below:

---

**Algorithm 1** NTN-Diff

---

**Input**: masked image $I$, binary mask $M$, text prompt $C$, null-text prompt $C_\emptyset$
**Output**: inpainted image $I'$

---

**I. Diffusion Process for Unmasked Regions**

 1: Extract the initial latent feature of the unmasked region $z_0^{gt} = E(I)$.
 2: **for** $t = 0$ to $T_{inv} - 1$ **do**
 3:     Compute $z_{t+1}^{gt}$ from $z_t^{gt}$;
 4: **end for**{DDIM inversion}

---

**II. Early Stage**

 1: Initialize $z_T^{un}, z_T^{text}, z_T^{in} \sim \mathcal{N}(0, 1)$.
 2: **for** $t = T$ to $\lambda T + 1$ **do**
 3:     Substitute the **unmasked regions** of $z_t^{un}$ with the corresponding counterpart of $z_t^{gt}$ via **Eq.3**;
 4:     Compute $z_{t-1}^{un}$ from $z_t^{un}$, conditioned on $C_\emptyset$ via **Eq.4**;
 5:     Substitute **low-frequency band** of $z_t^{text}$ with the corresponding counterpart of $z_t^{un}$ via **Eq.5–7**;
 6:     Compute $z_{t-1}^{text}$ from $z_t^{text}$, conditioned on $C$;
 7:     Substitute **mid-frequency band** of $z_t^{in}$ with the corresponding counterpart of $z_t^{text}$ via **Eq.8–10**;
 8:     Compute $z_{t-1}^{in}$ from $z_t^{in}$, conditioned on $C_\emptyset$;
 9: **end for**

---

**III. Late Stage**

 1: **for** $t = \lambda T$ to $1$ **do**
 2:     Substitute **unmasked regions** of $z_t^{in}$ with the corresponding counterpart of $z_t^{gt}$ via **Eq.11**;
 3:     Compute $z_{t-1}^{in}$ from $z_t^{in}$, conditioned on $C$;
 4: **end for**
 5: Output the final inpainted image $I' = D(z_0^{in})$.

---

# D    More Analysis of Computational Efficiency in the Sec.3.2 of the Mainbody

As indiacted in the mainbody, we further provide the inference time and VRAM consumption of our proposed NTN-Diff model, and compared it with several representative baselines, which are reported in Table. 5. Our inference time is comparable to **BrushNet** [12] and faster than recent methods such as **HDP** [25] and **PP** [49]. Moreover, our model consumes less VRAM than other text-guided inpainting models, **owing to the fact that our method performs low- and mid-frequency substitution across different branches after each denoising step**. The key idea is that, being decoupled from the resource-intensive denoising process itself, our approach introduces only negligible additional memory cost compared to the single denoising process in **BLD** [1].

Table 5: Inference time and VRAM consumption comparison among different models. NTN-Diff achieves a comparable inference time while exhibiting the lowest memory cost among all methods.

| Methods (Venue) | Inference Time (s) ↓ | VRAM Consumption (GB) ↓ |
|---|---|---|
| BLD [1] (TOG'23) | **4.78** | **2.11** |
| PP [49] (ECCV'24) | 7.24 | 4.09 |
| BrushNet [12] (ECCV'24) | 6.59 | 3.19 |
| HDP [25] (ICLR'25) | 10.57 | 11.75 |
| **NTN-Diff (Ours)** | 6.96 | **2.11** |

Table 6: Quantitative comparisons between NTN-Diff* and other diffusion-based inpainting models over EditBench for inpainting are shown, where all models use Stable Diffusion V1.5 as the baseline model. red and blue stand for the best and second best result. NTN-Diff* achieves the best result.

| Metrics | | Masked Region Preservation | | | | Text Alignment |
|---|---|---|---|---|---|---|
| **Models** | **Venue** | **PSNR**↑ | **MSE**$_{\times 10^3}$↓ | **LPIPS**$_{\times 10^3}$↓ | **SSIM**$_{\times 10^2}$↑ | **CLIP Score**↑ |
| **SDI** [32] | CVPR' 22 | 23.25 | 6.94 | 24.30 | 90.13 | 28.00 |
| **BLD** [1] | TOG' 23 | 20.89 | 10.93 | 31.90 | 85.09 | 28.62 |
| **CNI** [47] | ICCV' 23 | 12.71 | 69.42 | 159.71 | 79.16 | 28.16 |
| **CNI*** [47] | ICCV' 23 | 22.61 | 35.93 | 26.14 | 94.05 | 27.74 |
| **PP** [49] | ECCV' 24 | 23.34 | 20.12 | 24.12 | 91.49 | 27.80 |
| **BrushNet*** [12] | ECCV' 24 | 33.66 | 0.63 | 10.12 | 98.13 | 28.87 |
| **BrushEdit*** [17] | ArXiv' 24 | 32.97 | 0.70 | 7.24 | 98.60 | 29.62 |
| **HDP** [25] | ICLR' 25 | 23.07 | 6.70 | 24.32 | 92.56 | 28.34 |
| **NTN-Diff* (Ours)** | - | 36.83 | 0.28 | 5.39 | 99.46 | 29.72 |

[*] with blending operation of BrushNet

# E   Additional Quantitative and Qualitative analysis in the Sec.3.2 of the Mainbody

**Evaluation metric.** We adopt five metrics to evaluate the inpainted results below: Peak Signal-to-Noise Ratio (**PSNR**), structural similarity index (**SSIM**) [37] and Mean Squared Error (**MSE**) evaluate low-level pixel-wise differences between generated images and their ground truth counterparts. Additionally, Learned Perceptual Image Patch Similarity (**LPIPS**) [48] measures perceptual similarity by computing the distance between deep features extracted from a pre-trained neural network, offering a robust metric for assessing perceptual alignment; CLIP Similarity (**CLIP Score**) [38] measures text-image consistency by projecting both the generated images and their corresponding text prompts into a shared embedding space using the CLIP model. It then evaluates the similarity between their embeddings.

As indiacted in the mainbody, we further exhibit additional quantitative results by performing the experiments on the EditBench [35]; see Table.6. It is observed that **NTN-Diff** enjoys larger PSNR, SSIM and CLIP Score, together with smaller MSE and LPIPS than the competitors. Notably, **BrushEdit**[17] remains the large performance margins (at most, 3.86% for PSNR, 0.42% for MSE 1.85% for LPIPS, 0.86% for SSIM, and 0.1% for CLIP Score) compared to **NTN-Diff** in Table.6. We also present the quantitative results of **NTN-Diff** with the pixel-level blending operation of [12], named **NTN-Diff***, to preserve the unmasked regions, which demonstrates the ability to tame the hybrid frequency issue, the results further verifies the intuition in Sec.1 of the mainbody – *NTN-Diff can achieve the semantics consistency between mid-and-low frequency bands across masked and unmasked regions, while preserving unmasked regions.*

To shed more light on the advantages of our method, we further perform the visual analysis on the inpainted results with *the higher resolution*. Fig.9 delivers the following: due to the discrepancy between the diffusion process for unmasked regions substitution and the denoising process for masked region alignment upon text prompt. BLD [1] inevitably generate the content out of the masked regions (the first column of the Fig.9(a)). Note that, PP [49] and BrushNet [12] often exhibits the unreasonable inpainted result, *e.g.*, The extra woman(the 4th row of PP in Fig.9(a)) and the disappeared dinosaur (the 1st row of BrushNet in Fig.9(b)), owing to the unmasked regions are disrupted, thus fails to reconstruct the masked region as per text prompt, which attributes to the fact that *the low-frequency bands for both masked and unmasked regions are easily to be changed by text prompts, especially under the early stage of the text-guided denoising process with high-level noise. As opposed to that, mid-frequency band can better achieve the semantics consistency between masked and unmasked regions, while preserve its own frequency band well, than low-frequency band, owing to its robustness to the text prompt during the denoising process, while the previous arts fail to disentangle all frequency bands during the denoising process especially for its early stage with high-level noise* (see Sec.1 of the mainbody).

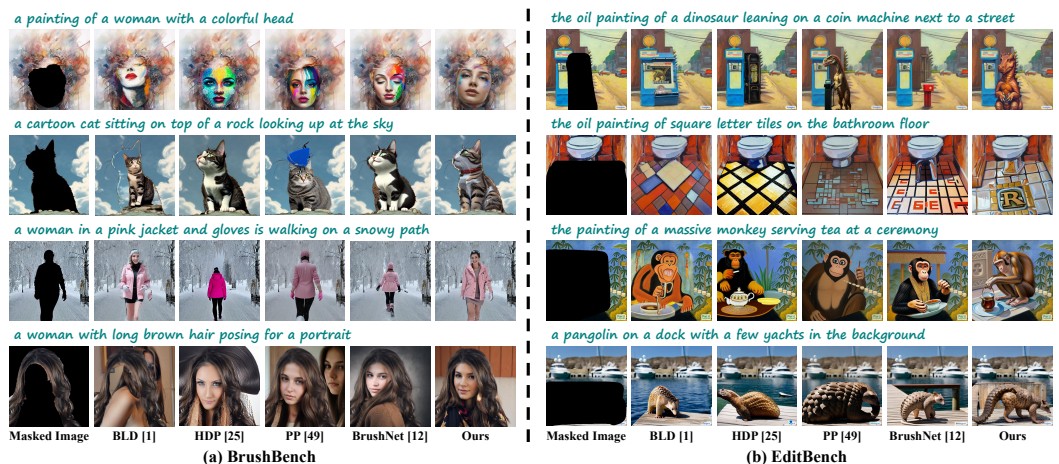

| a painting of a woman with a colorful head | the oil painting of a dinosaur leaning on a coin machine next to a street |
| a cartoon cat sitting on top of a rock looking up at the sky | the oil painting of square letter tiles on the bathroom floor |
| a woman in a pink jacket and gloves is walking on a snowy path | the painting of a massive monkey serving tea at a ceremony |
| a woman with long brown hair posing for a portrait | a pangolin on a dock with a few yachts in the background |

Masked Image   BLD [1]   HDP [25]   PP [49]   BrushNet [12]   Ours       Masked Image   BLD [1]   HDP [25]   PP [49]   BrushNet [12]   Ours

(a) BrushBench                       (b) EditBench

Figure 9: Comparison of the text-guided inpainted results with the state-of-the-arts on BrushBench [12] and EditBench [35]. NTN-Diff delivers the superior inpainted results over others, which can simultaneously preserve the unmasked regions while achieve the semantics consistency between unmasked and inpainted masked regions.

Table 7: Ablation studies on the impact of text and null-text prompts on low-and-mid frequency bands during denoising process: TTN-Diff, NTT-Diff and TTT-Diff for the early stage, based on the same text-guided denoising process (Sec.2.4 of the mainbody) for the late stage. **red** and **blue** stand for the best and second best result.

| Dataset | Metric | TTN-Diff | NTT-Diff | TTT-Diff | Ours(NTN-Diff) |
|---|---|---|---|---|---|
| **EditBench** | $\mathbf{IR}_{\times 10}\uparrow$ | 1.57 | **2.12** | 1.34 | **3.10** |
| | **PSNR**$\uparrow$ | 22.57 | **22.58** | 22.51 | **22.65** |
| | **LPIPS**$_{\times 10^3}\downarrow$ | 25.12 | **24.99** | 25.24 | **24.21** |
| | **CLIP Score**$\uparrow$ | 28.87 | **28.89** | 28.81 | **28.95** |
| **BrushBench** | $\mathbf{IR}_{\times 10}\uparrow$ | 9.61 | **10.32** | 9.22 | **11.12** |
| | **PSNR**$\uparrow$ | 28.06 | **28.08** | 28.01 | **28.10** |
| | **LPIPS**$_{\times 10^3}\downarrow$ | 44.63 | **44.55** | 44.88 | **44.09** |
| | **CLIP Score**$\uparrow$ | 25.94 | **25.97** | 25.89 | **26.09** |

# F   More Ablation Studies on the Impact of Text and Null-Text Prompts on Low-and-Mid Frequency Bands during Denoising Process

As mentioned in Sec.3.3.1 of the mainbody, due to page limitation, we further provide more ablation studies on the impact of text and null-text prompts on low-and-mid frequency bands during denoising process on BrushBench and EditBench datasets with three variants: **TTN-Diff**: replacing the first null-text denoising process with the text-guided denoising process; **NTT-Diff**: replacing the last null-text denoising process with the text-guided denoising process; **TTT-Diff**: replacing both the first and last null-text denoising processes with the text-guided denoising processes; Table.7 suggests that our **NTN-Diff** outperforms **NTT-Diff**, despite the denoised mid-frequency band well aligns with the text prompts especially for masked regions, while also encodes the information related to low-frequency band from the first null-text denoising process, which verifies that *the last null-text denoising process can achieve semantics consistency between mid-and-low frequency bands across masked and unmasked regions, by denoising the low-frequency band throughout the path to be semantically consistent to mid-frequency band, with no influence from text prompts*, which is in line with Sec.2.3.3 of the mainbody; **TTN-Diff** exhibits a substantial performance degradation, confirming that *the null-text denoising process conditioned on null-text prompt can avoid being influenced by text prompts even under the high-level noise, focusing primarily on low-frequency band,* hence validating the important the first null-text denoising process (Sec.2.3.1 of the mainbody). We illustrate the above intuitions in Fig.10.

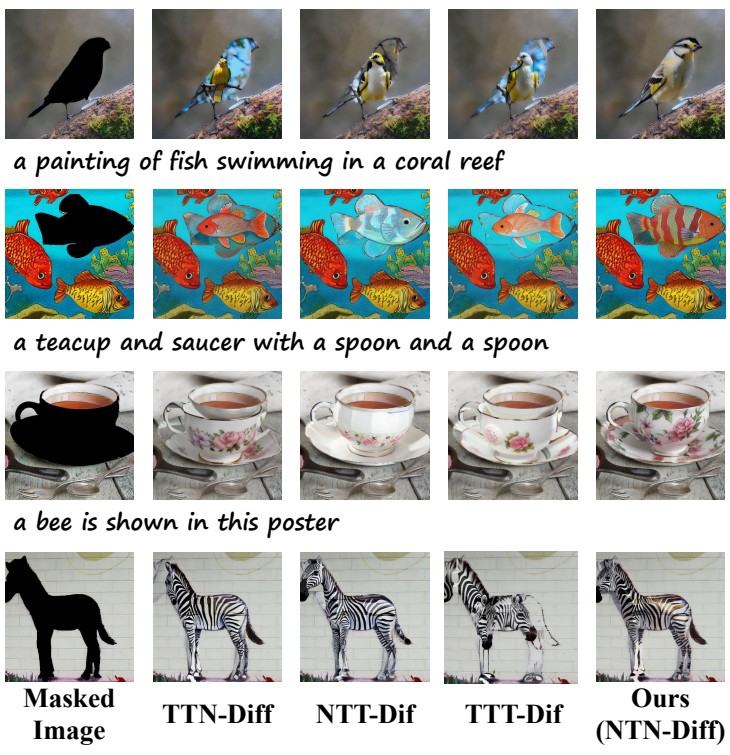

a bird with a yellow and white face sitting on a tree branch

a painting of fish swimming in a coral reef

a teacup and saucer with a spoon and a spoon

a bee is shown in this poster

| Masked Image | TTN-Diff | NTT-Dif | TTT-Dif | Ours (NTN-Diff) |

Figure 10: The inpainted output about the impact of text and null-text prompts on low-and-mid frequency bands during denoising process, with the first null-text denoising process (Sec.2.3.1 of the mainbody), the second text-guided denoising process (Sec.2.3.2 of the mainbody) and the last null-text denoising process (Sec.2.3.3 of the mainbody), NTN-Diff can achieve the better inpainted results than others.

## G  Additional Generation Results for the Ablation Study in Sec.3.3.2 of the mainbody

As mentioned in Sec.3.3.2 of the mainbody, due to page limitation, we further provide more visual results to analyse the parameter $\lambda$ in the denoising process (Sec.2.2 of the mainbody), which is utilized to divide the denoising process into *early* and *late* stages, demarcated by the critical step $\lambda T$. see Fig.11. When $\lambda = 0.9$, the performance is the worst with the shortest early stage, such as blue background in the 4th row. When $\lambda = 0.6$ with the balance of early and late stage, the best performance of image quality, unmasked region preservation and text alignment are achieved, such as the face of the cat in the first rows, confirming the rational that *the denoised low-frequency band for the masked regions is guided by the denoised mid-frequency band within the whole early stage, which is guided by text prompts for a few number of steps from text-guided late denoising process to avoid the large influence from text prompts to be inconsistent for unmasked regions, thus the desirable length of early stage can make the low-frequency band under the mid-frequency guidance for masked regions achieve the consistency to the substituted unmasked regions for ground truth*, which is consistent to Sec.2.4 of the mainbody.

## H  Limitations and Broader Impacts

**Limitations.** Unlike previous state-of-the-art methods that fine-tune Stable Diffusion (v1.5) on large-scale datasets for text-guided image inpainting, which typically require extensive time and computational resources for training, our method introduces a plug-and-play frequency-aware null-text-null diffusion framework. This framework substitutes mid-and-low frequency bands during the

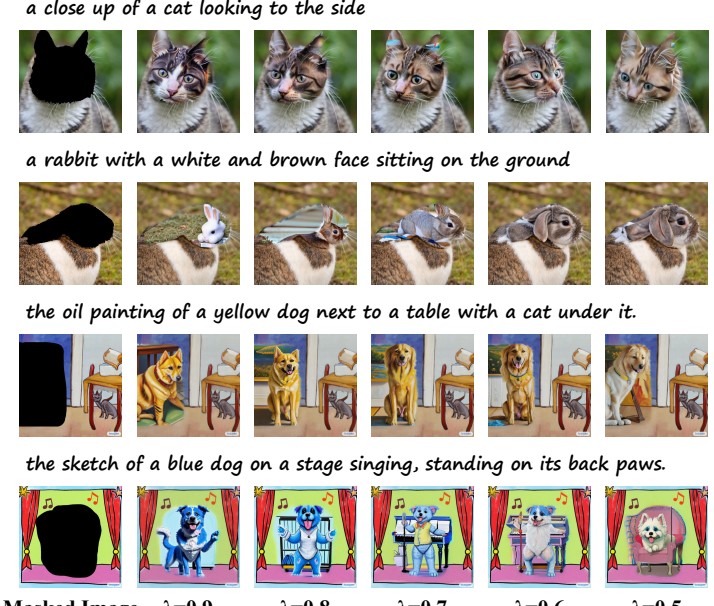

Figure 11: The inpainted output about hyperparameter sensitivity analysis of $\lambda$ for the length of the early and late stage, when $\lambda = 0.6$ with the balance of early and late stage, NTN-Diff can achieve the better inpainted results than others.

early stage of the denoising process and, as a result, involves three parallel null-text-null denoising processes in the early stage, resulting in a larger computational cost than single process. Nevertheless, it shares the same order of magnitude as the previous methods.

**Broader Impacts.** Our NTN-Diff can achieve the masked regions to be generated according to the text prompt, while preserve the unmasked regions, our method may raise certain ethical concerns. The inpainted image could potentially be misused in the creation of misleading information. We strongly advocate for the establishment of clear accountability in the use of such technologies, along with enhanced legal and technical oversight, to ensure they are applied responsibly.

