# OpenReview forum: "One Stone with Two Birds: A Null-Text-Null Frequency-Aware Diffusion Models for Text-Guided Image Inpainting"
_NeurIPS.cc/2025/Conference — NeurIPS 2025 poster_

### Official Review · Reviewer_7rxu · 2025-06-28

**Clarity:** 3
**Significance:** 2
**Originality:** 3
**Rating:** 5
**Confidence:** 4

**Summary:**

This paper addresses the problem of image inpainting within text-to-image (T2I) diffusion models. Specifically, the authors aim to tackle two key challenges: preserving the fidelity of unmasked regions and ensuring semantic consistency between the unmasked and inpainted (masked) regions. To this end, they propose a framework called Null-Text-Null Frequency-Aware Diffusion (NTN-Diff). The core idea is to decompose semantic consistency across masked and unmasked regions into frequency bands and handle them separately at different denoising stages. Experiments are conducted on the BruchBench and EditBench benchmarks, using Stable Diffusion v1.5 as the base model.

**Questions:**

Please see the weaknesses specified above.

**Ethical Concerns:**

["NO or VERY MINOR ethics concerns only"]

**Final Justification:**

The rebuttal answered my questions in terms of computational complexity and generalization ability to other diffusion based models.

**Limitations:**

The limitations are discussed in the appendix.

**Paper Formatting Concerns:**

There appear to be some noticeable formatting adjustments made to accommodate long content.

**Quality:**

3

**Strengths And Weaknesses:**

**S1:** The idea is well-motivated and logically structured. The core motivation to disentangle the varying influence of different frequency components across denoising stages is supported by both analytical insights and empirical evidence.

**S2:** The paper is well-written, with the key technical components clearly explained and easy to follow.

**S3:** The experimental results demonstrate consistent performance improvements over multiple baseline methods.


**W1:** The iterative sampling process is likely to extend the denoising procedure and introduce additional inference overhead. A discussion of the computational cost and a comparison with existing methods would provide a more complete picture of the proposed framework’s practicality.

**W2:** Since the proposed method relies on the structure of DDPMs, its generalizability to more recent T2I models, such as DiTs or flow-based models, may be limited. Moreover, the current experiments are conducted solely on Stable Diffusion v1.5.

Minor: There appear to be some noticeable formatting adjustments made to accommodate long text sequences.

---

> ### Author Rebuttal · Authors · 2025-07-30
>
> # To Reviewer 7rxu
> Thanks very much for your valuable comments that helped us significantly improve this submission. Below are detailed point-to-point responses about how we address your questions.
>
> ---
>
> # [W1]: Question on Computational Efficiency
>
> **Response**:Thanks, we kindly suggest you to refer to [Q1] from **Reviewer hGay**, to facilitate your reading, we made up the inference time and VRAM consumption of our model, comparing it with other baseline models, along with the detailed explanations below:
>
>
> | Methods (Venue)       | Inference Time (s) ↓ | VRAM Consumption (GB) ↓ |
> |----------------------|:------------------:|:----------------------:|
> | BLD [1] (TOG’ 23)  |       **4.78**         |        **2.11**            |
> | PP [2]  (ECCV’ 24)          |       7.24         |        4.09            |
> | BrushNet [3]  (ECCV’ 24)    |       6.59         |        3.19            |
> | HDP [4] (ICLR’ 25)         |      10.57         |       11.75            |
> | **NTN-Diff (Ours)** |       6.96         |        **2.11**            |
> |||
>
> * **Analysis of computational efficiency**
>
> *"Our inference time is comparable to BrushNet [3] and faster than recent methods such as HDP [4] and PP [2]. Moreover, our model consumes less VRAM than other text-guided inpainting models, **owing to the fact that our method performs low-and-mid frequency substitution across different branches after each denoising step**. The basic idea is to be decoupled from the resource-intensive denoising process itself, it incurs only negligible additional memory cost than the single denosing process BLD [1]."*
>
>
> ---
>
> # [W2]: Results on More T2I Models
>
> **Response**:Thanks for the valuable comments. We kindly clarify that we adopt **Stable Diffusion v1.5 (SD v1.5)** since most existing text-guided image inpainting methods fine-tune their models based on SD v1.5. To ensure a fair comparison, we mainly present the inpainted results based on SD v1.5 in the mainbody.
>
> Following your suggestion, we additionally conducted the inpainted results of other representative text-to-image (T2I) models, including the DiT-based method **HunyuanDiT** and **Stable Diffusion XL (SDXL)**, which is an upgraded version of SD v1.5. We report the quantitative results on **BrushBench**, covering both **inside** and **outside** inpainting tasks, along with detailed explanations, as shown below:
>
> | Method| $IR_{×10}$ ↑ | $HPS_{×10²}$ ↑ | CLIP Score ↑ |
> |---------------------|:-------:|:-----------:|:------------:|
> |BrushBench (Inside) - **SD v1.5**|9.61|27.52|26.90|
> |BrushBench (Inside) - **SDXL**| 11.85 |28.10 |27.92 |
> |BrushBench (Inside) - **HunyuanDiT**| 12.07| 28.36 | 28.06 |
> ||||||
> |BrushBench (Outside) - **SD v1.5**| 5.29 | 25.31  | 26.12 |
> |BrushBench (Outside) - **SDXL**|7.43 | 26.68 |26.84|
> |BrushBench (Outside) - **HunyuanDiT**| 7.69 | 26.82 |26.98|
> ||||||
>
> * **Analysis of the improvements achieved on powerful T2I models**
>
> *"We adopt the IR and HPS to evaluate the image generation quality, and use the CLIP score to assess the text-image alignment, which is in line with **Lines 270–277, Sec. 3.1 of the mainbody**. The results show that both HunyuanDiT and SDXL outperform SD v1.5 on BrushBench. This improvement can be attributed to their significantly larger parameter scales and training on more diverse and extensive datasets. As a result, in the latent space, the features of the unmasked regions retain more information after downsampling. This enables our frequency-aware null-text-null diffusion framework to exchange richer mid-and-low frequency components during the early stage of the denoising process, which is consistent to the motivation in **Sec. 2.3 of the mainbody**."*
>
> ---
>
> # [W3]: Writing and Presentation
>
> **Response**: Thanks for the comments, we will fix all your suggestions regarding writings issues, such as long sentences, small figure and typos etc.
>
>
> ---
> # References
>
> - [1] Avrahami O, Fried O, Lischinski D. Blended latent diffusion[J]. ACM transactions on graphics (TOG), 2023, 42(4): 1-11.
>
> - [2] Zhuang J, Zeng Y, Liu W, et al. A task is worth one word: Learning with task prompts for high-quality versatile image inpainting[C]//European Conference on Computer Vision. Cham: Springer Nature Switzerland, 2024: 195-211.
>
> - [3] Ju X, Liu X, Wang X, et al. Brushnet: A plug-and-play image inpainting model with decomposed dual-branch diffusion[C]//European Conference on Computer Vision. Cham: Springer Nature Switzerland, 2024: 150-168.
>
> - [4] Manukyan H, Sargsyan A, Atanyan B, et al. Hd-painter: high-resolution and prompt-faithful text-guided image inpainting with diffusion models[C]//The Thirteenth International Conference on Learning Representations. 2025.

---

> > ### Comment · Reviewer_7rxu · 2025-08-03
> >
> > I thank the authors for their rebuttal. It answered my questions, and I will raise my rating to 5.

---

> ### Author Response · Authors · 2025-08-03
>
> Dear Reviewer 7rxu:
>
> We are sincerely grateful to the  updated rating as you raised up and positive confirmation to our work, while largely benefited from your intuitive comments. Many thanks !

---

### Official Review · Reviewer_zpgQ · 2025-06-30

**Clarity:** 2
**Significance:** 2
**Originality:** 3
**Rating:** 4
**Confidence:** 4

**Summary:**

This paper proposes NTN-Diff, a frequency-aware diffusion model for text-guided image inpainting. The core idea is to improve consistency between masked and unmasked regions by aligning their low- and mid-frequency components during denoising. The method uses a three-stage early denoising process to disentangle and coordinate frequency bands, followed by standard text-guided refinement in the late stage.

**Questions:**

1. The authors note that “compared to the mid-frequency band, the high-frequency band is sparser and more easily overwhelmed by high-level noise in the early stages,” and thus focus primarily on the mid- and low-frequency bands. Does this imply that the proposed method is currently unable to handle or preserve high-frequency content in the un-masked regions? Given that high-frequency cues (e.g., edges and fine textures) are critical for human perception, this design choice seems somewhat counterintuitive. Could the authors provide an analysis or discussion comparing image quality with and without accurate high-frequency information? Such insights would be valuable for the community to better understand how different frequency bands contribute to perceptual quality in text-guided image generation.

2. Equations (6) and (9) suggest that the low-pass and mid-pass masks are predefined based on coordinate thresholds, and thus are data-independent. However, it is intuitive that the distribution of low-, mid-, and high-frequency components can vary significantly across different images. Would it not be more appropriate to adopt data-dependent frequency masks that adapt to the specific frequency characteristics of each input? Could the authors clarify this design decision, and potentially discuss whether a learned or adaptive masking strategy might further improve performance?

3. The paper lacks experimental evidence to support the necessity of the proposed three-stage denoising process in the early phase (null-text → text-guided → null-text), which likely incurs significant inference overhead. To better justify this design, it would be helpful for the authors to provide comparisons against simplified versions of the pipeline—such as single-stage or two-stage denoising—in terms of both performance (e.g., semantic consistency, fidelity) and computational efficiency (e.g., runtime, FLOPs). Quantifying how much performance degrades when reducing the number of stages would help clarify the necessity and cost-effectiveness of the current design.

4. The visualizations and SOTA comparisons are not particularly convincing. For instance, in Figure 5, several examples (e.g., the woman with a colorful headscarf) suggest that BrushNet already preserves the unmasked regions reasonably well. This observation seems to contradict the claim in the introduction that existing methods "fail to preserve unmasked regions." Given this discrepancy, could the authors clarify whether the metrics shown in Figure 1, particularly those based on CLIP distance, reliably reflect unmasked region preservation? Additional qualitative or user-based validation may be needed to support this point.

**Ethical Concerns:**

["NO or VERY MINOR ethics concerns only"]

**Final Justification:**

The authors have addressed all of my concerns. This is a solid paper. Although it didn’t particularly appeal to me, it indeed presents a complete piece of work. Therefore, I am raising my rating to borderline accept.

**Limitations:**

yes

**Quality:**

2

**Strengths And Weaknesses:**

Strengths:
1. The paper introduces a novel frequency-aware perspective for text-guided image inpainting. Specifically, it proposes to retain low- and mid-frequency components from the unmasked regions to guide the inpainting process. This spectrum-based formulation of guidance is intuitive and technically interesting.

2. This paper includes statistical observations on how different frequency bands behave during early-stage denoising, which may be helpful to the community.

Weaknesses:
1. While the method is grounded in frequency-domain intuition, the exploration and justification of frequency-related design choices lack depth.

2. The multi-stage denoising structure increases complexity, but its necessity and efficiency are not adequately quantified.

3. Some visual results do not clearly support the paper’s core claims, raising concerns about the reliability of the evaluation metrics used.

---

> ### Author Rebuttal · Authors · 2025-07-30
>
> # To Reviewer zpgQ
> Thanks very much for your valuable comments that helped us significantly improve this submission. Below are detailed point-to-point responses about how we address your questions.
>
> ---
> # [Q1]: Discussion on the High-Frequency Band
>
> **Response**: Thanks. As elaborated in **Remark 1 (Lines 202–207, Sec. 2.3.2) of the mainbody**, to ensure consistency between masked and unmasked regions, we introduce a concurrent null-text denoising process guided by the previously denoised mid-frequency components. This process helps to denoise the low-frequency band across both masked and unmasked regions. Meanwhile, the **high-frequency band** is also addressed in the final null-text denoising step of the early stage.
>
> Following your suggestions, we further conduct the ablation studies on exploiting the denoised high-frequency result from the second text-guided denoising process, to substitute the denoised high-frequency band from the last null-text denoising process, along with the detailed explanations, as shown below:
>
> | Method| $IR_{×10}$ ↑ | PSNR ↑ | $LPIPS_{×10³}$ ↓ | CLIP Score ↑ |
> |---------------------|:-------:|:------:|:-----------:|:------------:|
> |EditBench-$Case_{HF}$|2.36|22.49|25.03|28.63|
> |**EditBench-Ours**|**3.10**|**22.65**|**24.21**|**28.95**|
> |||
> |BrushBench-$Case_{HF}$|10.55|28.06|44.71|25.98|
> |**BrushBench-Ours**|**11.12**|**28.10**|**44.09**|**26.09**|
> |||
>
> * **Analysis of the impact of substituting the denoised high-frequency components**
>
> *"The results show that $Case_{HF}$ exhibits a substantial performance degradation (at most 23.87% for IR), confirming that **the high-frequency band is inherently sparser and more susceptible to being overwhelmed by high-level noise during early denoising, which significantly limits its effectiveness in guiding low-frequency reconstruction**, which is in line with **footnote in Sec. 1**, as well as **the Line 53 - 61 of mainbody**."*
>
> * We experimentally validate the denoising process as per several metrics to measure changes in the low-and-high frequency bands at different timesteps, we adopt two groups of metrics to analyze the frequency components of the denoising process below:
>
> | Timesteps | Mean Gray | Mean HSV-V | Mean Luma | Avg Gradient | Variance | LBP Variance |
> |-----------|:---------:|:----------:|:---------:|:------------:|:--------:|:------------:|
> |**0**|**120.9**|**136.0**|**120.9**|**59.01**|**3955.0**|**4.45**|
> |5|120.8|134.6|120.8|75.07|4098.3|3.71|
> |10|120.7|133.5|120.7|101.58|4378.9|3.25|
> |15|120.7|133.7|120.7|133.26|4790.8|3.07|
> |20|120.9|135.1|120.9|164.23|5274.8|3.07|
> |25|121.2|137.3|121.2|190.49|5743.9|3.15|
> |30|121.4|139.2|121.4|209.84|6099.5|3.25|
> |35|121.1|140.0|121.1|222.03|6291.1|3.32|
> |40|120.5|140.1|120.5|228.75|6363.9|3.35|
> |45|119.9|139.8|119.9|232.19|6394.8|3.38|
> |50|119.5|139.5|119.5|233.88|6419.5|3.40|
> ||
>
> * **Analysis of the high-and-low frequency in the denoising process**
>
> *The results show that low-frequency metrics (Mean Gray, HSV-V, Luma) remain nearly unchanged from the beginning, indicating early stabilization. In contrast, high-frequency metrics (Avg Gradient, Variance, LBP Variance) increase steadily in the late timesteps, **confirming that high-frequency details are refined gradually during denoising**, which is in line with **Lines 54–57, Sec. 1 of the mainbody**."*
>
> ---
> # [Q2]: Discussion on Adaptively Extracting the Low-and-Mid Frequency Bands
>
> **Response**: Thanks for your valuable comments. We kindly remind that we truly considered to **adaptively** extract the low-and-mid frequency bands from masked images based on **Eq.(7) and Eq.(10) of the mainbody, we copy that as follow:**
>
> * **Line 197 - 198, Eq.(7) of the mainbody**
>
> $$
> th_{lp} = λ_{lp}^{f} + λ^{r}_{lp} \cdot \frac{\|M\|_1}{HW}
> $$
>
> *where $λ_{lp}^{f}$ and $λ_{lp}^{r}$ are hyper-parameters. We set $th_{lp}$ to adaptively correlate with the ratio of unmasked regions, since the larger unmasked regions require more **low-frequency band** from the null-text denoising process to substitute the low-frequency band within the text-guided denoising process.*
> * **Line 219 - 220, Eq.(10) of the mainbody**
>
> $$
> th_{mp1} = λ_{mp1}^{f} - λ_{mp1}^{r} \cdot \frac{\|M\|_1}{HW}
> $$
>
> $$
> th_{mp2} = λ_{mp2}^{f} + λ_{mp2}^{r} \cdot \frac{\|M\|_1}{HW}
> $$
>
> *where $\lambda_{mp1}^{f}$, $\lambda_{mp1}^{r}$, $\lambda_{mp2}^{f}$, and $\lambda_{mp2}^{r}$ are hyper-parameters.  We also set $th_{mp1}$ and $th_{mp2}$ to correlate with the ratio of unmasked regions.  This is because larger nmasked regions require more **mid-frequency band** to encode the information that originates from the **low-frequency band** substituted during the first null-text denoising process.*
>
> We provide further ablation studies on the impact of adaptively extracting low-and-mid frequency bands in the **Line 70 - 89, Sec.C of the Appendix, i.e., Table. 2, we copy that below:**
>
> * **Table. 2 of the Appendix**
>
> |Method|  $IR_{×10}$ ↑ |   PSNR ↑  | $LPIPS_{×10³}$ ↓ | CLIP Score ↑ |
> | ----------------------- | :-------: | :-------: | :----------: | :----------: |
> |EditBench-CaseI|2.98|22.53|25.12|28.91|
> |EditBench-CaseII|3.02|22.50|25.20|28.87|
> |EditBench-CaseIII|3.07|22.62|24.55|28.93|
> |**EditBench-Ours**|**3.10**|**22.65**|**24.21**|**28.95**|
> |||
> |BrushBench-CaseI|10.04|28.08|44.18|25.91|
> |BrushBench-CaseII|10.10|**28.12**|44.25|25.89|
> |BrushBench-CaseIII|10.95|28.08|44.15|26.07|
> |**BrushBench-Ours**|**11.12**|28.10|**44.09**|**26.09**|
> |||
>
> * **Analysis of adaptively extracting the low-and-mid frequency bands**
>
> *"As mentioned in **Sec.2.3.2 and Sec.2.3.3 of the mainbody**, we set $th_{lp}$ in Eq.(7), and $th_{mp1}$ and $th_{mp2}$ in Eq.(10) to correlate with the ratio of unmasked regions. Due to page limitations, we provide further ablation studies on the impact of adaptively extracting low-and-mid frequency bands ... **Case III** shows substantial performance degradation, confirming *that larger unmasked regions demand more low-frequency information from the null-text denoising process to replace the low-frequency bands in the text-guided denoising process*, which is consistent with the findings in **Sec.2.3.2 of the mainbody**. These observations are also illustrated in Table 2."*
>
> ---
> # [Q3]: Discussion on the Performance and Computational Efficiency
> **Response**: Following your suggestions, we extend the experiments in **Table 2 and Line 300 - 313, Sec. 3.3.1 of the mainbody** to more results regarding the inference time and VRAM consumption during the early-stage denoising process, as shown below:
>
> * **Table. 2 of the mainbody**
>
> | Method              | $IR_{×10}$ ↑ | $LPIPS_{×10³}$ ↓ | CLIP Score ↑ | Inference Time (s) ↓ | VRAM Consumption (MB) ↓ |
> |---------------------|:------------:|:-----------------:|:-------------:|:----------------------:|:-------------------------:|
> |EditBench-CaseB|1.24(–60.00%)|24.82(–2.52%)|28.53(–1.45%)|**4.78**(–2.18s)|**2159.35**(–3.94MB)|
> |EditBench-CaseC|2.14(–30.97%)|25.24(–4.25%)|28.60(–1.21%)|5.51(–1.45s)|2163.17(–0.12MB)|
> |**EditBench-Ours**|**3.10**|**24.21**|**28.95**|6.96|2163.29|
> |||
> |BrushBench-CaseB|9.59(–13.77%)|47.08(–6.78%)|25.78(–1.19%)|**4.78**(–2.18s)|**2159.35**(–3.94MB)|
> |BrushBench-CaseC|10.02(–9.89%)|44.92(–1.88%)|26.03(–0.23%)|5.51(–1.45s)|2163.17(–0.12MB)|
> |**BrushBench-Ours**|**11.12**|**44.09**|**26.09**|6.96|2163.29|
> |||
>
> * **Analysis of the balance of performance and computational efficiency**
>
> *"The results demonstrates that our method significantly improves key performance metrics across both EditBench and BrushBench datasets. For example, IR increases by up to 44.86% and 10.99%, while LPIPS and CLIP Score also show consistent improvements, indicating better image quality and semantic alignment. Although inference time increases by 2.18s, this trade-off is fully justified given the substantial gains in generation quality and consistency. **Overall, the performance boost far outweighs the moderate increase in computational cost.** "*
>
> ---
> # [Q4]: Discussion on the Unmasked Region Preservation
>
> **Response**: Thanks. We kindly clarify that the CLIP-based metric shown in **Fig.1 of the mainbody** is used as a complementary metric, for measuring the preservation of unmasked regions. While CLIP distance primarily captures semantic-level similarity, it can help reveal whether the denoised image has undergone undesirable semantic shifts—including in the unmasked areas.
>
> To ensure a more comprehensive evaluation, we have included multiple metrics (*e.g.*, LPIPS, PSNR, SSIM and MSE). These jointly verify that our method better preserves the unmasked regions, which demonstrates the ability to tame the hybrid frequency issue, the results further verify the intuition in **Sec.1 of the mainbody** - NTN-Diff achieves the semantics consistency between mid-and-low frequency bands across masked and unmasked regions, while preserving unmasked regions. To facilitate the validation, **we copy the quantitative results** of unmasked region preservation between BrushNet [1] and ours **in Table 1 of the mainbody and Appendix below**:
>
>
> * **Table. 1 of the mainbody**
>
> | Method | PSNR ↑ | $MSE_{×10³}$ ↓ | $LPIPS_{×10³}$ ↓ |
> |--------------------------|:------:|:---------:|:-----------:|
> |BrushBench(Insides)-BrushNet [1]|21.65|9.31|48.28|
> |**BrushBench(Insides)-NTN-Diff(Ours)**|**23.51**|**6.50**|**40.79**|
> |BrushBench(Outsides)-BrushNet [1]|18.06|22.86|15.08|
> |**BrushBench(Outsides)-NTN-Diff(Ours)**|**18.47**|**20.44**|**14.46**|
> |||
>
> * **Table. 1 of the Appendix**
>
> | Method| PSNR ↑ | $SSIM_{×10²}$↑ | $MSE_{×10³}$ ↓ | $LPIPS_{×10³}$ ↓ |
> |--------------------------|:------:|:------:|:---------:|:-----------:|
> |EditBench-BrushNet [1]|33.66|98.13|0.63|10.12|
> |**EditBench-NTN-Diff(Ours)**|**36.83**|**99.46**|**0.28**|**5.39**|
> |||
>
> ---
> # References
>
> - [1] Ju X, Liu X, et al. Brushnet: A plug-and-play image inpainting model with decomposed dual-branch diffusion[C]//European Conference on Computer Vision. Cham: Springer Nature Switzerland, 2024: 150-168.

---

> > ### Comment · Reviewer_zpgQ · 2025-08-03
> >
> > Thank you for the detailed and thoughtful response. I appreciate the authors' efforts in addressing the concerns raised in the initial review. In particular, the newly added discussion on the high-frequency band effectively clarifies the underlying mechanisms and resolves my primary concern. I will raise my rating accordingly and hope the paper will be accepted.

---

> ### Author Response · Authors · 2025-08-03
>
> Dear Reviewer zpgQ:
>
> Many Thanks for your positive confirmation for our rebuttal and our work; undoubtedly, your constructive comments significantly help improve our work, where we are also deeply grateful to the updated rating as you raised up, along with your best wishes.

---

### Official Review · Reviewer_uzVF · 2025-07-02

**Clarity:** 1
**Significance:** 3
**Originality:** 3
**Rating:** 4
**Confidence:** 3

**Summary:**

The robustness of diffusion models to text prompting varies during the course of the denoising process. At the start of the reverse process (when the image is mostly noise), the reversal is very susceptible perturbations from the text prompting (low frequency changes are common). In order to fix this, the authors propose using several reverse processes that are each coupled together. The main reversal is “null-text” at large noise levels, and text-guided at low noise levels. A separate but coupled text-guided reversal is used to fill in the mid-frequencies of the null-text process at large noise levels, and a third null-text process is used to fill in the low frequencies of the text-guided reversal used above. At low noise levels, the forward process from the masked ground truth itself is coupled into the reversal.

Note: Overall I recommend accept because this seems like a decent effort to solve an interesting problem with a novel idea. I hesitate to say this because I think this paper needs an important major rewrite, albeit for superficial reasons.

**Questions:**

1. In Figure 1, what is the red dashed line? It says “Diffusion result” in the figure, but the caption says that it plots the difference between the ground truth and the text prompt in CLIP latent space. Why is that changing? Is Figure 1(d) just a completely independent figure that depicts a comparison with other methods?
2. It is very difficult to see Figure 2. It seems like there are depictions of the low frequency band and mid-frequency band of the latent space. Does that mean that you pass the latent representation through a low/band pass filter? Can this be explained formally somewhere? This seems like an interesting idea but I have never seen it before and I feel a reader could use a more gentle introduction to this topic.
3. Is there some formal way of saying that when you do a diffusion in pixel space the lower frequency components get decided first? It seems like a simple model of this should be possible - for instance the mean brightness of the noise you sample at the beginning might be correlated with the mean brightness of the resulting image and is determined already at the start of the process. It seems natural that in pixel space you cannot change lower frequency components as easily as higher frequency components.
4. Is it possible to just use the low frequency and high frequency components of a null-text model with mid-frequency components of a text-guided model? In other words, why are there two null-text processes?

**Ethical Concerns:**

["NO or VERY MINOR ethics concerns only"]

**Final Justification:**

Thank you for your response. I will keep my score.

**Limitations:**

Yes

**Quality:**

3

**Strengths And Weaknesses:**

Strengths:
This is a good idea. If indeed it is the case that we would like to avoid text conditioning at large noise levels, this seems like a reasonable way to do it, while still maintaining the mid-frequency learned by the text prompting. Basically I think the idea is to use a text-conditioned reversal and a null-text reversal and couple them in such a way as to get the conditioning from the text-guided process and the robustness of the null-text process.

Weaknesses:
I found the writing and presentation difficult to follow. I dont mean to be unnecessarily picky about this, but it should be easy enough to use simpler writing choices, especially with today’s tools.
1. What does “One stone with two birds” have to do with the paper?
2. There is a >70 word sentence in the abstract that can likely be broken into at least 3 smaller sentences.
3. There is no pseudo code, or any other such consolidated description of the algorithm. What I expressed in the summary is based off of staring at Figure 3 for a while.
4. All of the figures are too small, or contain titles, axis descriptions that are difficult to see even when you zoom in.
5. Line 33, preserve -> preserving. Possibly many more examples of this.

---

> ### Author Rebuttal · Authors · 2025-07-30
>
> # To Reviewer uzVF
> Thank you very much for your valuable comments that helped us significantly improve this submission. Below are detailed point-to-point responses about how we address your questions.
>
> ---
> # [W1]: The Meaning of “One stone with two birds”
>
> **Response**: Thanks for your valuable comments. We would like to clarify that, as mentioned in **Line 32 - 52, Sec.1 of the mainbody**, *the longstanding challenges of text-guided image inpainting lie in **the preservation for unmasked regions**, while **achieving the semantics consistency between unmasked and masked regions** as inpainted. Previous arts failed to address both of them, always with either of them to be remedied.* In this paper, we propose a null-text-null frequency-aware diffusion models, named **NTN-Diff**, for text-guided image inpainting, **to simultaneously address two challenges** of **unmasked regions preservation**, along with its **semantics consistency with inpainted masked regions**. **Hence, we title one stone with two birds.**
>
>
> ---
> # [W2]: Pseudo Code of the Algorithm
>
> **Response**: As you suggested, we summarize the pseudo code of the algorithm below, which is consistent to our code in **Sec.G of the Appendix** for reproducibility.
>
> > ### **Algorithm: NTN-Diff**
> >
> > ---
> > **Input**: masked image $I$, binary mask $M$, text prompt $C$, null-text prompt $C_{∅}$
> > **Output**: inpainted image $I'$
> >
> > ---
> > #### **I. Diffusion Process for Unmasked Regions**
> > 01: Extract the initial latent feature of the unmasked region $z_{0}^{gt} = E(I)$.
> > 02: **for** $t = 0$ to $T_{inv} - 1$ **do**
> > 03:  Compute $z_{t+1}^{gt}$ from $z_{t}^{gt}$;
> > 04: **end for** {DDIM inversion}
> >
> > ---
> > #### **II. Early Stage**
> > 05: Initialize $z_{T}^{un}$, $z_{T}^{text}$, $z_{T}^{mid} \sim N(0,1)$.
> > 06: **for** $t = T$ to $\lambda T + 1$ **do**
> > 07:  Substitute the **unmasked regions** of $z_{t}^{un}$ with the corresponding counterpart of $z_{t}^{gt}$ via **Eq.3**;
> > 08:  Compute $z_{t-1}^{un}$ from $z_{t}^{un}$, conditioned on $C_{∅}$ via **Eq.4**;
> > 09:  Substitute **low-frequency band** of $z_{t}^{text}$ with the corresponding counterpart of $z_{t}^{un}$ via **Eq.5–7**;
> > 10:  Compute $z_{t-1}^{text}$ from $z_{t}^{text}$, conditioned on $C$;
> > 11:  Substitute **mid-frequency band** of $z_{t}^{mid}$ with the corresponding counterpart of $z_{t}^{text}$ via **Eq.8–10**;
> > 12:  Compute $z_{t-1}^{mid}$ from $z_{t}^{mid}$, conditioned on $C_{∅}$;
> > 13: **end for**
> >
> > ---
> > #### **III. Late Stage**
> > 14: **for** $t = \lambda T$ to $1$ **do**
> > 15:  Substitute **unmasked regions** of $z_{t}^{mid}$ with the corresponding counterpart of $z_{t}^{gt}$ via **Eq.11**;
> > 16:  Compute $z_{t-1}^{mid}$ from $z_{t}^{mid}$, conditioned on $C$;
> > 17: **end for**
> > 18: Output the final inpainted image $I' = D(z_{0}^{mid})$.
>
> ---
> # [W3]: Writing and Presentation
>
> **Response**: Thanks for the comments, we will fix all your suggestions regarding writings issues, such as long sentences, small figure and typos etc.
>
> ---
> # [Q1]: The Explanation of Fig.1
>
> **Response**: Thanks for the valuable comments. To better understand **red dashed lines**, we kindly clarify that t-SNE visualization of the CLIP latent space is adopted to illustrate the behavior of the text-guided denoising process in **Fig.1 (a–c) of the mainbody**, which aims to examine whether our method can **preserve the unmasked regions** and **achieve semantic consistency between masked and unmasked regions** across timesteps in the denoising process.
>
> * Instead of computing the direct distance between the noise-free ground truth and the text prompt, **red dashed line** in the Fig.1 highlights the trace the denoising trajectory of the ground truth at different timesteps and connect them to serve as a reference path.
>
> * Besides, we compare our NTN-Diff (**Fig.1(c)**) with state-of-the-arts for text-guided inpainting: The BLD result in **Fig.1(d)** corresponds to the type represented in **Fig.1(a)**, while the BrushNet results in **Fig.1(d)** align more closely with the type in **Fig.1(b)**, which provides qualitative support for our analysis.
>
> ---
> # [Q2]: The Explanation of Fig.2
>
> **Response**: Thanks the comments. We visualize the low- and mid-frequency bands in the latent space during the denoising process to better illustrate their differences.
>
> * At each timestep, we apply the 2D Discrete Cosine Transform to the latent representation and use the masks defined in **Eq. (6) (Line 195 - 196) and Eq. (9) (Line 216 - 217) of the mainbody** to extract the corresponding low- and mid-frequency components.
>
> * We then apply the 2D Inverse Discrete Cosine Transform to project these frequency-filtered components back into the latent space, yielding the mid-frequency latent representation and the low-frequency latent representation.
>
> * To further facilitate visual comparison, we compute the corresponding timestep-0 representations by inverting the noise addition step defined in **Eq. (1) (Line 114 - 115) of the mainbody**—i.e., applying the reverse operation of the noise injection without running the full denoising process, to facilitate the observation on the influence of each frequency band on the final output in a more isolated and interpretable manner.
>
> * We then decode these latent representations into the image space via the pre-trained VAE decoder for visualization.
>
> ---
> # [Q3]: Changes in Low-and-High Frequency Components During Denoising Process
>
> **Response**: As you suggested, we conducted the experiments to verify the conclusion in **Lines 54–57, Sec. 1 of the mainbody**: *the low-frequency band fluctuates more significantly during the early stage of the denoising process with high-level noise than the mid-and-high frequency bands*. This observation is consistent with the fact that diffusion models typically recover the low-frequency components first and progressively refine the mid- and high-frequency details.
>
> Motivated by the above, we experimentally validate the denoising process as per several metrics to measure changes in the low-and-high frequency bands at different timesteps, as reported below:
>
> | Timesteps | Mean Gray | Mean HSV-V | Mean Luma | Avg Gradient | Variance | LBP Variance |
> |-----------|:---------:|:----------:|:---------:|:------------:|:--------:|:------------:|
> |**0**|**120.9**|**136.0**|**120.9**|**59.01**|**3955.0**|**4.45**|
> |5|120.8|134.6|120.8|75.07|4098.3|3.71|
> |10|120.7|133.5|120.7|101.58|4378.9|3.25|
> |15|120.7|133.7|120.7|133.26|4790.8|3.07|
> |20|120.9|135.1|120.9|164.23|5274.8|3.07|
> |25|121.2|137.3|121.2|190.49|5743.9|3.15|
> |30|121.4|139.2|121.4|209.84|6099.5|3.25|
> |35|121.1|140.0|121.1|222.03|6291.1|3.32|
> |40|120.5|140.1|120.5|228.75|6363.9|3.35|
> |45|119.9|139.8|119.9|232.19|6394.8|3.38|
> |50|119.5|139.5|119.5|233.88|6419.5|3.40|
> |||
>
> * **Analysis of the high-and-low frequency in the denoising process**
>
> "*We adopt two groups of metrics to analyze the frequency components of the denoising process:*
>
> * *For low-frequency information, we adopt **Mean Gray**, **Mean HSV-V**, and **Mean Luma**, which primarily capture the global brightness and luminance structure of the denoised image. These metrics help evaluate how well the low-frequency bands are maintained throughout the denoising process.*
>
> * *For high-frequency information, we use **Average Gradient**, **Variance**, and **LBP Variance**, which are sensitive to edge sharpness, local contrast, and texture complexity, respectively. These metrics reflect the preservation and recovery of fine details during denoising.*
>
> *The results show that low-frequency metrics (Mean Gray, HSV-V, Luma) remain nearly unchanged from the beginning, indicating early stabilization. In contrast, high-frequency metrics (Avg Gradient, Variance, LBP Variance) increase steadily in the late timesteps, **confirming that high-frequency details are refined gradually during denoising**, which is in line with **Lines 54–57, Sec. 1 of the mainbody**."*
>
> ---
> # [Q4]: Discussion on the Second Null-Text Denoising Process of the Early Stage
>
> **Response**: Thanks. as mentioned in **Line 202 - 207, Sec.2.3.2 of the mainbody**. We mildly clarify that *the low-frequency band across both masked and unmasked regions are preserved even under text prompts owing to the substitutions from **the first null-text denoising process**, with only mid-frequency aligned with text prompts for both regions, hence still failed to achieve the consistency between masked and unmasked regions. To address this, we propose another concurrent **null-text denoising process**, guided by above denoised mid-level frequency, to help denoise the low-frequency band for both regions, especially for masked regions*,  To verify that, we also conduct the ablation studies in **Line 301 - 313, Sec.3.3.1 of the mainbody, **i.e.**, Table. 2, we copy that below:**
>
>
> * **Table. 2 of the mainbody**
>
> |Method| $IR_{×10}$ ↑ | PSNR ↑ | $LPIPS_{×10³}$ ↓ | CLIP Score ↑ |
> |---------------------|:-------:|:------:|:-----------:|:------------:|
> | EditBench - Case C  | 2.14    | 22.53  | 25.24       | 28.60        |
> | EditBench - Ours   | **3.10**| **22.65** | **24.21**  | **28.95**    |
> ||||||
> | BrushBench - Case C | 10.02   | 28.06  | 44.92       | 26.03        |
> | BrushBench - Ours  | **11.12**| **28.10**| **44.09**  | **26.09**    |
> ||||||
>
>
> * **Analysis of the second null-text denoising process**
>
> *“...Case C: removing the null-text denoising process (Sec.2.3.3); ... Case C exhibits a substantial performance degradation (at most 44.8%), confirming that only mid-frequency aligned with text prompts for masked and unmasked regions, still failed to achieve the consistency across the whole regions, which is in line with Sec.2.3.3 and illustrated in Fig.6.”*

---

> > ### Comment · Reviewer_uzVF · 2025-08-04
> >
> > Thank you for your clarifications. I will keep my score.
> >
> > I thought the phrase "one stone with two birds" was a play of some kind. In the event that it is not, please note that the phrase is "two birds with one stone", meaning you solve two problems with one solution.

---

> ### Author Response · Authors · 2025-08-04
>
> Dear Reviewer uzVF:
>
> We truly agree with your thoughtful suggestions; while being grateful to your positive confirmation to our rebuttal, consistent positive rating and efforts for the improvement dropped to our work.
>
> Many thanks!

---

### Official Review · Reviewer_hGay · 2025-07-03

**Clarity:** 3
**Significance:** 2
**Originality:** 3
**Rating:** 5
**Confidence:** 4

**Summary:**

This paper aims to address a core contradiction in text-guided image inpainting: the need to generate semantically consistent content in the masked region according to the text prompt while perfectly preserving the original appearance of the unmasked region. The authors observe that previous methods struggle to balance these two requirements because different frequency components of an image have varying robustness to text prompts during the denoising process of diffusion models. Specifically, low-frequency components (such as color and lighting) are easily influenced by text prompts and can change, leading to damage in the unmasked region; while mid-frequency components (such as structure and layout) are more stable against text prompts and better suited for guiding semantic alignment.​
Based on this finding, the paper proposes a new method called NTN-Diff. The core idea of this method is to decouple and process image information in the frequency domain. It divides the denoising process into early and late stages. In the early stage, NTN-Diff meticulously processes different frequencies through three parallel denoising sub-processes:​
A null-text process, used to preserve low-frequency information in the unmasked region without text interference.​
A text-guided process, used to generate content aligned with the text in the masked region, while its low-frequency part is replaced by the result of the first process to initially protect the background.
Another null-text process, utilizing the mid-frequency information already aligned with text from the second process as guidance, to generate consistent low-frequency information, thereby addressing the semantic gap between masked and unmasked regions.
In the late stage, the model performs the final text-guided denoising while forcibly restoring the unmasked region at each step to ensure it remains unchanged. Through this complex, multi-stage, frequency-specific processing pipeline, NTN-Diff successfully achieves both protection of the unmasked region and semantic consistency between the generated region and the background.

**Questions:**

1. Question on Computational Efficiency

Your proposed NTN-Diff method runs three denoising processes in parallel during the early stage, which appears to incur significant computational overhead. To better assess the practicality of this approach, could you provide comparative data on inference time and VRAM consumption against major baseline methods (such as BrushNet)? This data is crucial for understanding the trade-off between performance improvement and resource consumption in your method.

2. Question on Robustness to Extreme Mask Ratios

The adaptive threshold for frequency-domain masking (Eq. 7 & 10) is correlated with the area ratio of the masked region. How does the method perform when processing very small or very large masked regions? Does a critical point exist where the effectiveness of the frequency-domain separation strategy diminishes? For instance, when the masked region is extremely large (i.e., the unmasked region is very small), is the low-frequency information from the unmasked region still sufficient to effectively guide the overall inpainting process?

3. Regarding the experiments on EditBench

Your paper only reports the results of your proposed method on this benchmark. The results of other methods such as BrushNet are not included. We strongly recommend supplementing this section with comparative performance data.

**Ethical Concerns:**

["NO or VERY MINOR ethics concerns only"]

**Final Justification:**

The authors' rebuttal partially resolved my concerns. After reviewing all feedback, I will raise my rating to 5.

**Limitations:**

yes

**Paper Formatting Concerns:**

The paper's formatting basically meets the requirements.

**Quality:**

3

**Strengths And Weaknesses:**

**Strengths**

1. Originality & Significance

The core contribution of this paper lies in providing a novel perspective—frequency-domain analysis—for text-guided image inpainting. Identifying the distinct behaviors of low- and mid-frequency components during the diffusion denoising process, and establishing this as the theoretical foundation for resolving the "content generation vs. background preservation" conflict, constitutes a very novel and insightful perspective.

2. Quality

  Technical Depth: The method is elegantly designed and technically sound. NTN-Diff is not a simple model but a meticulously crafted multi-stage, multi-process workflow.

  Strong Results: The authors evaluated their method on two mainstream benchmarks (BrushBench, EditBench) using six distinct metrics covering image quality, region preservation, and text alignment. NTN-Diff surpasses existing SOTA methods on BrushBench. Ablation studies demonstrate the effectiveness of each component within its complex design. Furthermore, sensitivity analysis of the key hyperparameter λ reflects the rigor of the research.

3. Clarity

The introduction clearly articulates the problem and challenges. Figures 1 and 2 play a crucial role in explaining the root cause of the problem and the motivation behind the method, providing excellent intuition. Despite the inherent complexity of the method, the authors make it understandable through step-by-step descriptions and flowcharts.


**Weaknesses**
1. Quality

  Computational Complexity: The most significant potential drawback is computational complexity. The requirement to run three independent denoising processes in parallel during the early stage could result in substantially increased inference time and VRAM consumption compared to single-process baselines (potentially up to threefold). The paper lacks quantitative analysis and comparison of this computational overhead, making it difficult to assess the method's practical feasibility.
  Potential for Design Simplification: While ablation studies confirm the utility of each component, whether this complex three-process architecture represents the only or optimal solution remains debatable.

2. Originality

The originality of this work manifests more in the clever integration and sophisticated application of existing ideas (e.g., frequency-domain decomposition, text-free inversion) than in proposing fundamentally new theories or techniques. However, this is only a minor weakness, as the integration itself is highly non-trivial and demonstrably effective.

---

> ### Author Rebuttal · Authors · 2025-07-30
>
> # To Reviewer hGay
> Thank you very much for your valuable comments that helped us significantly improve this submission. Below are detailed point-to-point responses about how we address your questions.
>
> ---
> # [Q1]: Question on Computational Efficiency
>
> **Response**: Thanks for your valuable comments. As per your suggestion, we made up the
> inference time and Peak VRAM consumption of our model, comparing it with other baseline models, along with the detailed explanations, as shown below:
>
> | Methods (Venue)       | Inference Time (s) ↓ | VRAM Consumption (GB) ↓ |
> |----------------------|:------------------:|:----------------------:|
> | BLD [1] (TOG’ 23)  |       **4.78**         |        **2.11**            |
> | PP [2]  (ECCV’ 24)          |       7.24         |        4.09            |
> | BrushNet [3]  (ECCV’ 24)    |       6.59         |        3.19            |
> | HDP [4] (ICLR’ 25)         |      10.57         |       11.75            |
> | **NTN-Diff (Ours)** |       6.96         |        **2.11**            |
> |||
>
> * **Analysis of computational efficiency**
>
> *"Our inference time is comparable to BrushNet [3] and faster than recent methods such as HDP [4] and PP [2]. Moreover, our model consumes less VRAM than other text-guided inpainting models, **owing to the fact that our method performs low-and-mid frequency substitution across different branches after each denoising step**. The basic idea is to be decoupled from the resource-intensive denoising process itself, it incurs only negligible additional memory cost than the single denosing process BLD [1]."*
>
>
> ---
> # [Q2]: Question on Robustness to Extreme Mask Ratios
>
> **Response**: Thanks for your valuable comments. According to **Line 246- 254, Remark 3 of the mainbody**, We kindly clarify that *due to the denoised mid-frequency band encodes information from the low-frequency band substituted from the null-text denoising process for ground truth including both masked and unmasked regions, **hence the denoised mid-frequency band not only aligns with the text prompts for both regions, but also achieving the consistency to the substituted unmasked regions for ground truth***,
>
> Based on the above, large masked regions make it easier to keep consistency between masked and unmasked regions. Our method adaptively extracts low-frequency bands. **So when the masked region is very large, only a small amount of low-frequency information is needed to maintain consistency**.
>
> We also provide quantitative comparisons on BrushBench to support this claim. As indicated in **Lines 261–265, Sec. 3.1 of the mainbody**, BrushBench refines the task by considering two specific scenarios: segmentation mask inside-inpainting and segmentation mask outside-inpainting. The outside-inpainting setting focuses on cases with large masked ratios, as shown in **Table. 1 (Outside-Inpainting) of the mainbody, we copy that as follow**:
>
> * **Table. 1 (Outside-Inpainting) of the mainbody**
>
> |Method (Venue)| $IR_{×10}$ ↑ | $HPS_{×10²}$ ↑ |  PSNR ↑  | $MSE_{×10³}$ ↓ | $LPIPS_{×10³}$ ↓ | CLIP Score ↑ |
> |--------------------------|:-------:|:------------:|:------:|:---------:|:-----------:|:------------:|
> | BLD [1] (TOG’ 23)              | 7.81    | 26.77        | 15.85  | 35.86     | 21.40       | 26.73        |
> | CNI [6] (ICCV’ 23)             | 9.26    | 27.68        | 11.91  | 83.03     | 58.16       | 27.29        |
> | PP [2] (ECCV’ 24)              | 7.45    | 28.01        | 18.04  | 31.78     | 15.13       | 26.72        |
> | BrushNet [3] (ECCV’ 24)        | 10.82| 28.02 | 18.06 | 22.86  | 15.08  | 27.33    |
> | HDP [4] (ICLR’ 25)             | 9.66    | 27.79        | 18.03  | 22.99     | 15.22       | 26.96        |
> | **NTN-Diff (Ours)**            | **11.54** | **28.22** | **18.47** | **20.44** | **14.46**   | **27.54**    |
> |||
>
> * **Analysis on the quantitative results of the large mask**
>
> *"The quantitative results summarize our findings below: **NTN-Diff** enjoys larger IR, HPS v2, PSNR and CLIP Score, together with smaller MSE and LPIPS than the competitors. Notably, BrushNet remains the large performance margins (at most 6.65% for IR, 0.71% for HPS, 2.27% for PSNR, 10.59% for MSE, 4.11% for LPIPS and 0.77% for CLIP Score) compared to **NTN-Diff**, which demonstrates the ability to tame the hybrid frequency issue, the results further verifies the intuition in **Sec.1 of the mainbody** -- **NTN-Diff** achieves the semantics consistency between mid-and-low frequency bands across masked and unmasked regions, while preserving the unmasked regions. We also illustrate the above intuitions in the **last two rows of Fig.1 (d) and the last row of Fig. 5(a) in the mainbody** "*
>
> ---
> # [Q3]: Regarding the experiments on EditBench
>
> **Response**: Thanks.We mildly remind you that we do present a comparison of text-guided inpainting results with state-of-the-art methods on **EditBench** in **the right part of Fig. 5 in the mainbody**. In addition, due to the page limitation for mainbody, we include further quantitative comparisons on **EditBench** with state-of-the-art approaches in the **Line 45-54, Sec. B of the Appendix**, *i.e.*, **Table. 1**, as indicated by **Line 298, Sec. 3.2 of the mainbody, we copy that as follow:**
>
>
> * **Table. 1 of the Appendix**
>
> |      Method (Venue)       | $IR_{×10}$ ↑ | $HPS_{×10²}$ ↑ |  PSNR ↑  | $MSE_{×10³}$ ↓ | $LPIPS_{×10³}$ ↓ | $SSIM_{×10²}$ ↑ | CLIP Score ↑ |
> |-------------------------|:-------:|:------------:|:--------:|:---------:|:-----------:|:----------:|:------------:|
> | SDI [5] (CVPR' 22)            |  1.86   |    24.24     |  23.25   |   6.94    |    24.30    |   90.13    |    28.00     |
> | BLD [1] (TOG' 23)             |  0.90   |    23.81     |  20.89   |  10.93    |    31.90    |   85.09    |    28.62     |
> | CNI [6] (ICCV' 23)            |  0.90   |    23.79     |  22.61   |  35.93    |    26.14    |   94.05    |    27.74     |
> | PP [2] (ECCV' 24)             |  1.24   |    24.50     |  23.34   |  20.12    |    24.12    |   91.49    |    27.80     |
> | BrushNet [3] (ECCV' 24)       |  4.46   |    25.24     |  33.66   |   0.63    |    10.12    |   98.13    |    28.87     |
> | BrushEdit [7] (ArXiv' 24)     |   -     |      -       |  32.97   |   0.70    |     7.24    |   98.60    |    29.62     |
> | HDP [4] (ICLR' 25)            |  1.74   |    24.20     |  23.07   |   6.70    |    24.32    |   92.56    |    28.34     |
> | **NTN-Diff (Ours)**          |**4.61**|**25.83**| **36.83**| **0.28**  |  **5.39**   | **99.46**  |  **29.72**   |
> |||
>
> * **Analysis of the quantitative results on EditBench**
>
> *"In particular, we further exhibit additional quantitative results by performing the experiments on the **EditBench**; It is observed that **NTN-Diff** enjoys larger PSNR, SSIM and CLIP Score, together with smaller MSE and LPIPS than the competitors. Notably, BrushEdit remains the large performance margins (at most, 3.86% for PSNR, 0.42% for MSE, 1.85% for LPIPS, 0.86% for SSIM, and 0.1% for CLIP Score) compared to **NTN-Diff** in table. The results further verifies the intuition in **Sec.1 of the mainbody** -- **NTN-Diff** can achieve the semantics consistency between mid-and-low frequency bands across masked and unmasked regions, while preserving unmasked regions."*
>
>
> ---
> # References
>
> - [1] Avrahami O, Fried O, Lischinski D. Blended latent diffusion[J]. ACM transactions on graphics (TOG), 2023, 42(4): 1-11.
>
> - [2] Zhuang J, Zeng Y, Liu W, et al. A task is worth one word: Learning with task prompts for high-quality versatile image inpainting[C]//European Conference on Computer Vision. Cham: Springer Nature Switzerland, 2024: 195-211.
>
> - [3] Ju X, Liu X, Wang X, et al. Brushnet: A plug-and-play image inpainting model with decomposed dual-branch diffusion[C]//European Conference on Computer Vision. Cham: Springer Nature Switzerland, 2024: 150-168.
>
> - [4] Manukyan H, Sargsyan A, Atanyan B, et al. Hd-painter: high-resolution and prompt-faithful text-guided image inpainting with diffusion models[C]//The Thirteenth International Conference on Learning Representations. 2025.
>
> - [5] Rombach R, Blattmann A, Lorenz D, et al. High-resolution image synthesis with latent diffusion models[C]//Proceedings of the IEEE/CVF conference on computer vision and pattern recognition. 2022: 10684-10695.
>
> - [6] Zhang L, Rao A, Agrawala M. Adding conditional control to text-to-image diffusion models[C]//Proceedings of the IEEE/CVF international conference on computer vision. 2023: 3836-3847.
>
> - [7] Li Y, Bian Y, Ju X, et al. Brushedit: All-in-one image inpainting and editing[J]. arXiv preprint arXiv:2412.10316, 2024.

---

> > ### Comment · Reviewer_hGay · 2025-08-08
> >
> > I thank the authors for their rebuttal, which partially resolved my concerns. After reviewing all feedback, I will raise my rating to 5.

---

> ### Author Response · Authors · 2025-08-08
>
> Dear Reviewer hGay,
>
> We are sincerely grateful to your positive confirmation and promoted rating.
>
> Many thanks for your great guidance for our work !

---

### Comment · Area_Chair_y579 · 2025-08-03

Dear Reviewers,

The author has provided a rebuttal response to each of your review comments. Please read all other comments and the author's responses carefully, and actively communicate with the authors if you have further questions or require answers from the author. You can engage in an open exchange with the authors.

Please post your response as soon as possible, so there is time for back and forth discussion with the authors.

Thanks, Your AC.

---

### Note · Authors · 2025-08-14

We are deeply grateful to AC and reviewers for the huge efforts in reviewing our paper.

All reviewers basically appreciate the strengths of  our paper as per their comments, especially for ***the novelty of our idea***, ***technical intuition*** and ***depth***, as well as ***the superior experimental results***.
- The reviewers pointed out some thoughtful questions, which are well clarified and validated by *positive confirmation from all reviewers*; *as we mildly remind all reviewers that **most of the answers are clearly available in the mainbody and Appendix of our manuscript***.

  To name the typical ones below:
  - **hGay** suggested conducting experiments on large masks and *EditBench*. We ***clarify*** and ***copy*** the results from **Lines 261-265, Sec. 3.1 and Table 1 of mainbody** and **Lines 45-54, Sec. B and Table 1 of Appendix**.
  - **uzVF** suggested discussing the second null-text denoising process at the early stage. We ***clarify*** and ***copy*** the ablation studies from **Lines 301-313 in Sec.3.3.1 and Table 2 of mainbody**.
  - **zpgQ** suggested discussing the adaptive extraction of low-and-mid frequency bands and unmasked region preservation. We ***clarify*** and ***copy*** the ablation studies from **Lines 70-89 in Sec. C, Table 1 and 2 of Appendix**, as well as **Eqs.(7)(10) and Table 1 in mainbody**.

   *All reviewers positively agree with our clarification.*
- As suggested by reviewers, we further provide more consistent experimental results with our current version, which are summarized below:
  - **hGay**, **zpgQ**, and **7rxu** suggested the computational efficiency. We made up the experiments and detailed explanations, please refer to our rebuttal to **Q1** of **hGay**, **Q3** of **zpgQ** and **W1** of **7rxu**.
  - **uzVF** suggested the pseudo-code of the algorithm and conducting experiments on the changes in low-and-high frequency. We made up the pseudo-code and the experiments, please refer to our rebuttal to **W2** and **Q3** of **uzVF**.
  - **7rxu** suggested experiments on more T2I models. We made up the experiments of other T2I models, please refer to our rebuttal to **W2** of **7rxu**.
- All four reviewers are basically satisfied with our rebuttal: *three reviewers response with promoted rating, with another one keeping the rating.*

Finally, ***we are sincerely grateful to AC with the timely reminder for the discussions.*** Hopefully, our remarks above can help facilitate the final assessment.

Many Thanks!

---

### Decision · Program_Chairs · 2025-09-17

**Decision:**

Accept (poster)

**Comment:**

After a comprehensive review of all reviewers' comments, we have decided to accept this paper. The paper's strengths, including its novelty, technical intuition and depth, as well as superior experimental results, have been generally appreciated by all reviewers.​

Regarding the thoughtful questions raised by the reviewers, the authors have provided clear clarifications and validations, which have received positive confirmation from all reviewers. It is noted that most of the answers were already available in the mainbody and Appendix of the manuscript, as gently reminded. Specific examples include clarifications on experiments related to large masks and EditBench, discussions on the second null-text denoising process at the early stage, and explanations on the adaptive extraction of low-and-mid frequency bands and unmasked region preservation, all of which have been addressed with references to relevant sections and tables.​
Furthermore, the authors have supplemented more consistent experimental results as suggested by the reviewers, covering aspects such as computational efficiency, algorithm pseudo-code, experiments on changes in low-and-high frequency, and experiments on more T2I models. These supplements have been well-received, with three reviewers upgrading their ratings and the remaining one maintaining their initial rating.​

We sincerely appreciate the authors' efforts in addressing the reviewers' concerns. This paper is thus recommended for acceptance.